# Unraveling Drought Tolerance and Sensitivity in Coffee Genotypes: Insights from Seed Traits, Germination, and Growth-Physiological Responses

Habtamu Chekol [1,*] , Yimegnu Bezuayehu [1], Bikila Warkineh [1] , Tesfaye Shimber [2],
Agnieszka Mierek-Adamska [3] , Grażyna B. Dąbrowska [3] and Asfaw Degu [1]

[1] Department of Plant Biology and Biodiversity Management, College of Natural and Computational Sciences, Addis Ababa University, Addis Ababa 3434, Ethiopia; bikila.warkineh@aau.edu.et (B.W.); asfaw.degu@aau.edu.et (A.D.)
[2] Ethiopian Institute of Agricultural Research, Addis Ababa 2003, Ethiopia; gessesetesfaye@yahoo.com
[3] Department of Genetics, Faculty of Biological and Veterinary Sciences, Nicolaus Copernicus University in Toruń, Lwowska 1, 87-100 Toruń, Poland; mierek_adamska@umk.pl (A.M.-A.); browsk@umk.pl (G.B.D.)
* Correspondence: habtamu.chekol@aau.edu.et

**Abstract:** The coffee plant is highly susceptible to drought, and different genotypes exhibit varying degrees of tolerance to low soil moisture. The goal of this work was to explore the interrelation between seed traits and germination events, growth patterns, and physiological responses of coffee genotypes, aiming to identify significant associations that may facilitate the selection of coffee genotypes exhibiting enhanced drought tolerance and yield potential. Two consecutive experiments were conducted to examine the impact of these factors. In the first experiment, germination performance was examined for three groups of coffee genotypes: relatively tolerant (*Ca*74140, *Ca*74112, and *Ca*74110), moderately sensitive (*Ca*74158, *Ca*74165, and *Ca*J-21), and sensitive (*Ca*754, *Ca*J-19, and *Ca*Geisha). The subsequent experiment focused on the growth and physiological responses of two relatively tolerant (*Ca*74110 and *Ca*74112) and two sensitive (*Ca*J-19 and *Ca*754) genotypes under drought stress condition. The relatively tolerant genotypes showed quicker and more complete germination compared to other groups. This was associated with higher moisture content, higher seed surface area to volume ratio, and higher coefficient of velocity of germination, coefficient of variation of germination time, and germination index. Additionally, the relatively tolerant genotypes showed higher seedling vigor. The results of the second experiment demonstrated superior growth performance in relative tolerant genotypes compared to the sensitive groups. Young coffee plants belonging to relatively tolerant genotypes exhibited higher growth performance than the sensitive genotypes, with a net assimilation rate strongly correlated to relative water content, leaf number, stomatal conductance, and chlorophyll-a. In addition, a strong correlation was exhibited between the growth of young coffee plants and the surface area to volume ratio of the seeds, as well as the germination percentage. The seedling vigor index showed a strong correlation with net assimilation rate, chlorophyll content, seedling growth, and cell membrane stability. Furthermore, principal component analysis illustrated distinct clustering of genotypes based on their germination and growth-physiological performance. Overall, the findings of this study suggest that seed traits, germination, and post-germination events are integral factors in determining drought tolerance and sensitivity, as well as the growth and physiological responses of adult coffee plants.

**Keywords:** Arabica coffee; drought; genotype; seed; germination; moisture content; seedling vigor; gas exchange; cell membrane stability





## 1. Introduction

*Coffea arabica* L. is the most widely cultivated commercial species, accounting for over 70% of the world's coffee production [1,2]. It is believed that the south and southwest of

Ethiopia are the center of origin and genetic diversity for *C. arabica* [3]. Coffee plants thrive best in areas where an altitude ranges between 1600 and 2800 m, rainfall is high, humidity is between 50% and 80%, light intensity is moderate, and slightly acidic soil is present [4,5]. Ethiopia is the leading Arabica coffee producer in Africa and the tenth-largest exporter worldwide, producing an average of ~471,000 tons per year with a yield of 0.71 tons/ha [6]. Ethiopian coffee is highly sought-after for its superior quality and organic nature [6–8].

Unfortunately, coffee production is significantly affected by drought events, with a large portion of the world's coffee being cultivated in drought-prone areas where the use of irrigation is uncommon [4]. Ethiopia is particularly challenged by recurrent drought stress due to increased temperatures and heightened air evaporative demand, which can cause a decrease in soil water availability [2]. Drought stress has a detrimental effect on coffee growth and is the major constraint in bean production and yields [2,3].

The propagation of *C. arabica* plants is usually done through their seeds, with the germination process initiating vegetative growth [9]. Healthy and properly stored seeds will germinate easily when the external and internal conditions necessary for germination are suitable [10]. Soil moisture has a profound impact on the coffee seed germination process and seedling emergence. Giorgini and Campos [11] stated that during drought periods, the imbibition process usually takes more time and delays before starting the adjustment of the seed's osmotic potential. As drought stress intensifies, the lack of imbibition and germination increases, which leads to poor germination, restricted radical and hypocotyl development, abnormal seedling, and poor plant establishment [12].

DaMatta et al. [13] reported that drought stress inhibits embryo, radical, and hypocotyls development, and affects shoot elongation and root growth. As a result, coffee genotypes that have the potential to store more seed moisture content imbibe water fast and have vigorous root development, can have successful seedling formation, which is an indication of tolerance to drought stress, and could easily avoid drought stress [10,13]. The effect of soil moisture on the activation of the embryo and the subsequent radicle development may substantially differ among coffee genotypes [14,15]. During the process of imbibition, hydrophilic molecules (-OH, -NH$_2$, -COOH, etc.) accumulate beneath the hard external layer of the coffee endosperm, drawing in water molecules. This causes a build-up of turgor potential within the seed, but further expansion is inhibited by an opposing mechanical force of the surrounding endosperm [16,17]. The water molecules initiate the mobilization of endogenous gibberellic acids towards the soft internal endosperm region, leading to the synthesis of hydrolyzing enzymes (endo-β-mannanase, cellulase, amylase, and protease) to breakdown the endosperm cell wall surrounding the embryo and create space for the embryo expansion and elongation, and weakening of the endosperm cap leading to the development of coffee seed protuberance [16,17]. Subsequently, the stored food reserves (carbohydrates, proteins, and lipids) break down into simpler biomolecules, such as simple sugars, amino acids, and fatty acids [18–20]. These simpler biomolecules then move toward the growing embryo, where they become metabolically active in the developing tissue [21,22].

Uniformity in seed germination and seedling vigor is essential for the successful establishment of commercial crops. However, coffee seeds are naturally characterized by asynchronous and slow germination [23]. This slow germination is caused by the loss of germination capacity and other related factors. In tropical rain-fed areas of arid and semiarid regions, soil moisture is the primary factor determining seed germination [13]. Additionally, the efficiency of seed germination among the coffee species and genotypes is influenced by the permeability of the endosperm (hard external and soft internal layer), temperatures, air moisture, seed moisture, seed damage, and other factors [10,20].

Drought stress is not only detrimental to the germination process but can also have a lasting impact on the growth, development, and yield of coffee plants [24]. At seedling and adult stages, the impact of drought stress begins at a cellular level and goes to the whole plant system [25–27]. Growth is established through cell division, cell growth, and differentiation, and low turgor pressure greatly limits the mitosis process and decreases

cellular division and further development [28]. If the severity of drought stress continues, it may even collapse the whole plant system [29]. Consequently, it influences early-stage developments (suppression of coleoptile, shoot, root length, etc.) [30], distorting osmotic balance and morphological changes [31], disrupting physiological activities [27,32], inhibiting biochemical properties [33], promoting oxidative stress, and even further affecting signal transduction, transcription, and translation factors, which are later accompanied by gene expression changes and the damage–repair process [34–37]. Hence, developing and screening coffee genotypes capable of withstanding drought stress and producing high yields is of utmost importance [26,38].

In Ethiopia, drought stress-associated research on coffee has mainly focused on water use, fertilizer application, agronomical practices, and yields. The current study aims to investigate the connection between drought tolerance and the sensitivity of seed traits, germination, and post-germination events, and their influence on the growth and physiology of young coffee plants. Based on this, morphological and developmental changes during coffee seed germination among seeds of genotypes that differ in their tolerance to drought stress, as well as the key germination-indicating factors associated with this process under drought stress conditions, were studied. Additionally, the impact of drought stress on growth performances, water relations, gas exchange, pigments, and cell membrane stability of young seedlings of some selected coffee genotypes were studied. Ultimately, this study is intended to assist in developing and/or screening genotypes that can withstand drought stress and produce high coffee yields.

## 2. Materials and Methods

### 2.1. Study Site

Both the shade-house and greenhouse experiments were conducted at the College of Natural and Computational Sciences of Addis Ababa University. The germination studies were conducted in the shade-house (40% shade level) within a poly-propagator wooden box (5 m × 1 m × 1 m, sand layered covered with polyethylene plastic sheet, which was maintained at a temperature of 26 °C, relative humidity of 55%, and a photon flux density of $345 \pm 16$ µmol m$^{-2}$s$^{-1}$), whereas the growth and physiological studies of adult coffee genotypes were conducted in a greenhouse (with the mean temperature of 24.5 °C, humidity of 50–70%, and photon flux density of $850 \pm 13$ µmol m$^{-2}$s$^{-1}$, with 12 h light/12 h dark photoperiod).

### 2.2. Plant Material

Nine *C. arabica* L. (*Ca*) genotypes obtained from Jimma Agricultural Research Center (JARC) were used in this study (Figure S1). These genotypes were selected based on their drought tolerance: relatively tolerant (*Ca*74140, *Ca*74112, and *Ca*74110), moderately sensitive (*Ca*74158, *Ca*74165, and *Ca*J-21), and sensitive (*Ca*754, *Ca*J-19, and *Ca*Geisha) [4]. Mature and healthy coffee berries were carefully handpicked from robust and healthy plants. The outer pericarp and mesocarp layers of the seeds were meticulously removed, leaving behind the endosperm (seed) along with the endocarp and spermoderm. Subsequently, these prepared seeds were kept inside zip lock plastic bags and stored under controlled refrigeration conditions at 4 °C with a relative humidity set below 40% until the initiation of the germination experiment. Pre-germination parameters (Tables S1 and S2) such as seed length (Sl, mm) and seed width (Sw, mm) (using a digital caliper to the nearest 0.01 mm), and seed fresh (Fw, g) and seed dry weight (Dw, g) (using balance to an accuracy of 0.01 g) were measured. The seed's initial moisture content (Mc), surface area (SA), and volume (SV) were calculated using the following equations [39]:

Seed initial moisture content (%):

$$Mc = \left( \frac{F_w - D_w}{D_w} \right) \times 100 \tag{1}$$

where $F_w$ is the fresh weight and $D_w$ is the dry weight. *Mc* was calculated based on the loss in weight as a percentage of the dry weight of the seeds.

Surface area (mm$^2$):

$$SA = Sl \times Sw \tag{2}$$

where *Sl* is seed length and *Sw* is seed width.

Seed volume (cm$^3$):

$$SV = \frac{\pi Sl Sw^2}{6} \tag{3}$$

where *Sl* is seed length and *Sw* is seed width, assuming that the width is equal to thickness.

### 2.3. Germination and Post-Germination Experiment

The sand was sieved with a 2 mm sieve, thoroughly washed with tap water, and sterilized in an oven at around 180 °C for 3 h. After cooling to room temperature, it was spread thinly on germination plastic trays (5 cm deep) to allow for radicle development [36,37]. Prior to sowing, the coffee seeds were retrieved from storage and subjected to a series of preparatory steps. The endocarp was promptly removed, and the spermoderm was thoroughly washed (Figure S2). Subsequently, the endosperm underwent a sterilization process by immersing it in a solution composed of 95% ethanol and 30% hydrogen peroxide in a 1:1 ratio (*v:v*) for a duration of 10 min. Next, seeds were imbibed in cold-distilled water for 12 h [21,38].

The germination experiment was started in August 2021. Coffee seed surfaces are highly susceptible to pathogens. To mitigate the risk of contamination and coffee seed infections, it is strongly advised to limit the number of seeds to 15–20 per tray [2]. In line with this recommendation, 20 seeds from each genotype were sown at a depth of 1 cm in plastic trays, each with three replications. The trays were placed in a randomized complete block design within a poly-propagator (Figure S3) [2]. To prevent spatial effects, the trays were moved randomly within the poly-propagator once a week. The trays were irrigated daily until the seedlings reached the "matchstick" size (before the emergence of cotyledons) (Figure S4). For the growth and physiological studies, the germinants were then transplanted into pots containing composite soil. Daily trial management and observation of germination were performed until the radical of each seed reached 2 mm in length, signifying the completion of the germination.

Germination parameters (Tables S1 and S3) such as germination percentage (GP) [40], the mean germination time (MGT) [41], coefficient of variation of germination time (CV$_t$) [42,43], coefficient of the velocity of germination (CVG) [44], germination index (GI) [45], germination rate index (GRI) [46], the uncertainty of germination process (U) [47,48], synchrony of germination process (Z) [49], mean daily germination percent (MDG) [50], peak value for germination (Pv) [50], and germination value (Gv) [51], were calculated based on the following formulas:

Germination percentage (%):

$$GP = \left( \sum\nolimits_{i=1}^{k} n_i / N \right) \times 100 \tag{4}$$

where $n_i$ is the number of seeds germinated in the $i^{th}$ time, $N$ is the number of all seeds that completed germination, and $k$ is the total number of time intervals.

Mean germination time (day):

$$MGT = \frac{\sum_{i=1}^{k} n_i t_i}{\sum_{i=1}^{k} n_i} \tag{5}$$

where $t_i$ is the time from the start of the experiment to the $i^{th}$ interval, $n_i$ is the number of seeds germinated in the $i^{th}$ time interval (not the accumulated number, but the number corresponding to the $i^{th}$ interval), and $k$ is the total number of time intervals.

Coefficient of variation of germination time (%):

$$CVt = \frac{S_t}{MGT} \tag{6}$$

where $S_t$ is standard deviation of germination time and $MGT$ is the mean germination time.

Coefficient of velocity of germination (%):

$$CVG = \frac{\sum_{i=1}^{k} n_i t_i}{\sum_{i=1}^{k} n_i} \times 100 \tag{7}$$

Germination index (day):

$$GI = \sum_{i=1}^{k} n_i / t_i \tag{8}$$

where $n_i$ is the number of seeds germinated in the $i^{\text{th}}$ time and $t_i$ is the time needed for seeds to germinate at the $i^{\text{th}}$ count.

Germination rate index (%/day):

$$GRI = \frac{G_1}{1} + \frac{G_2}{2} + \ldots \frac{G_n}{n} \tag{9}$$

where $G_1$ is the germination percentage on the first day after sowing and $G_2$ is the germination percentage on the second day after sowing, $G_n$ is the germination percentage on the $n$ day after sowing.

Uncertainty of germination process (degree of uncertainty) (bit):

$$U = \sum_{i=1}^{k} f_i \log_2 f_i \tag{10}$$

where $f_i = \frac{n_i}{\sum_{i=1}^{k} n_i}$, $f_i$ is relative frequency of germination, $n_i$ is the number of seeds germinated in the $i^{\text{th}}$ time interval, and $k$ is the total number of time intervals.

Synchrony of germination process (degree of overlapping):

$$Z = \frac{\sum_{i=1}^{k} C_{cni,2}}{C_{\sum ni,2}} \tag{11}$$

where $C_{ni,2} = n_i(n_i - 1)/2$, $C_{ni,2}$ is the partial combination of the two germinated seeds from among $n_i$, from the number of seeds germinated on the $i^{\text{th}}$ time interval, $C_{\sum ni,2}$ is the partial combination of the two germinated seeds from among the total number of seeds germinated at the final count, assuming that all seeds that germinated did so simultaneously.

Mean daily germination percent (%):

$$MDG = \frac{GP}{T_n} \tag{12}$$

where $GP$ is the final germination percentage and $T_n$ is the total number of intervals required for final germination.

Peak value (% day$^{-1}$):

$$Pv = \max\left(\frac{G_1}{t_1}, \frac{G_2}{t_2}, \ldots \frac{G_k}{t_k}\right) \tag{13}$$

where $t_i$ is the time from the start of the germination to the $i^{\text{th}}$ interval, $G_i$ is the cumulative germination percentage in the $i^{\text{th}}$ time interval, and k is the total number of time intervals.

Germination value (%$^2$day$^{-1}$):

$$Gv = MDG \times Pv \tag{14}$$

where *MDG* is the mean daily germination and *Pv* is the peak value.

Post-germination parameters, seedlings at 90 days after germination, such as root length (RL), shoot length (SdL), the ratio of root/shoot length (R/S$^r$), numbers of lateral roots (Rn) were also measured. The vigor index (VI), at 90 days of growth, was calculated based on the following formula [52]:

Vigor index:

$$VI = (SdL + RL) \times GP \tag{15}$$

where *SdL* is the mean shoot length, *RL* is the mean root length, and *GP* is the germination percentage.

Seedling morphological changes during germination and post-germination phases were also photographed using SonyAlphaA7RIV (Sony Group Corporation, Bangkok, Thailand) and observed under a Leica MZ8 microscope (Leica Microsystems Ltd., Heerbrugg, Switzerland) at 100 dpi resolution.

### 2.4. Growth and Physiological Experiment

To examine the growth and physiological response of coffee genotypes during the adult stage, a follow-up experiment was conducted on selected relatively tolerant (*Ca*74110 and *Ca*74112) and sensitive (*Ca*754 and *Ca*J-19) groups. After germination, when the first pair of leaves appeared, seedlings were transplanted into 5 L plastic pots with an aluminum foil covering at the side and top to prevent excessive heat build-up and evaporative loss. Each pot was filled with a 4 L potting mix of topsoil, compost, and sand in a ratio of 2:1:1 and contained a perforated bottom (9 mm diameter holes) for drainage. Coffee seedlings were then managed in a greenhouse, as reported by WCR [2], until the end of the experiment (Figure S5). After developing 7–8 pairs of leaves (around 150 days of age), each genotype was subjected to two different conditions: a well-watered (WW) and drought-stressed (WS) condition. In the WW condition, the plants were irrigated to field capacity every 3–4 days and served as the control group. In the WS condition, the seedlings were initially fully irrigated at field water capacity before the experiments began and then deprived of water until the end of the experiment. We employed a completely randomized block design to create a factorial arrangement of 4 × 2 (four genotypes and two water application treatments) combinations. For non-destructive parameters, including stem height, stem diameter, leaf number, and leaf area, we utilized 15 replications for each genotype. On the other hand, destructive parameters were evaluated at 10-day intervals, involving three coffee plants for each measurement time point.

#### 2.4.1. Plant Growth and Physiological Measurements

To evaluate the growth performance in response to drought stress, at 10 day intervals until the end of the experiment (for around 60 days), measurements such as stem height (SH) and stem collar diameter (SD, at the collar of the plants) were measured using a meter scale and caliper (500-197-30, Mitutoyo group, Kanagawa, Japan), respectively, while leaf number (LN) was counted manually. The leaf area (LA) was calculated as proposed by Tavares-Junior et al. [53].

$$LA = cE \tag{16}$$

where *E* is an estimated area (*E* = length × width), and *c* is the coefficient index (*c* = 0.99927).

At the end of the experiment (plants at around 210 days of age), the biomass of seedlings was assessed according to Vertregt and De Vries [54]. The plants from the two treatments were uprooted between 9:00 and 11:00 a.m. and the roots were carefully excavated and cleaned with tap water over a 0.5 mm screen sieve. The fresh weights (root fresh mass—RFM, stem fresh mass—SFM, leaf fresh mass—LFM, and total fresh mass—TFM) were measured on a weighing balance (Sartorius, Germany), the tap root length (RL, line intersect method), root number (RN), and root volume (RV, using the water-displacement method in a graduated cylinder) were measured, counted, and calculated.

$$RV = V_1 - V_2 \tag{17}$$

where *RV* is root volume, $V_1$ is the water volume after submerging the coffee roots into the graduated cylinder, and $V_2$ is the volume of the water in the graduated cylinder before submerging the coffee roots.

Further, the oven-dry mass (70 °C for 24 h) of the root (RDM), stem (SDM), leaf (LDM), and total dry mass (TDM) of the coffee genotypes was measured.

### 2.4.2. Stem Water Potential

The stem water potential ($\Psi_w$) was measured, at 9:00–11:00 a.m., using a Scholander pressure chamber (Soil Moisture Equipment Corp., Santa Barbara, CA, USA), at 10 days intervals till the end of the experiment (for around 60 days). Due to the small size of the leaf petioles, only the stem water potential of each genotype was measured. The stems were excised using a sharp blade and placed into the pressure chamber. The chamber was pressurized using a nitrogen tank, and $\Psi_w$ was recorded when the initial xylem sap was emerging from the cut end of the stem.

### 2.4.3. Leaf Relative Water Content

Based on the works of Barrs and Weatherley [55], at 10 day intervals until the end of the experiment (for around 60 days), relative water content (RWC) from representative leaves of the coffee genotypes was determined following the parameters:

$$RWC = \frac{(FW - DW)}{(TW - DW)} \times 100 \tag{18}$$

where *FW* is leaf fresh weight, *DW* is leaf dry weight, and *TW* is leaf turgid (re-saturated) weight.

The fresh weight of the root and shoot of the genotypes was measured, and for the determination of turgid weight, samples were soaked in distilled water for about 2 h at room temperature (20–22 °C) and weighed. Furthermore, for the determination of dry weight, the samples were dried to a constant weight at 70 °C.

### 2.4.4. Gas Exchange Measurements

Instantaneous gas exchange measurements were periodically measured out at 10 day intervals until the end of the experiment (for around 60 days). The rate of net carbon assimilation ($A_{net}$, μmol $CO_2$ m$^{-2}$ s$^{-1}$), stomatal conductance (Gs, mol $H_2O$ m$^{-2}$ s$^{-1}$), and transpiration rate (E, mmol $H_2O$ m$^{-2}$ s$^{-1}$) were collected using an open gas exchange system LI-6400 (LI-COR, Lincoln, Nebraska, USA) adjusted at 1000 μmol m$^{-2}$s$^{-1}$ photosynthetic photon flux density, 400 μmol $CO_2$ mol$^{-1}$ air reference $CO_2$ concentration, and 500 μmol s$^{-1}$ flow rates. The measurements were conducted between 9:00 and 11:00 a.m., on a young and fully expanded leaf.

### 2.5. Content of Photosynthetic Pigments

Following the protocols of Lichtenthaler [56], for the analysis of pigment (chlorophylls), healthy and fully expanded leaf discs from the same leaves used for gas exchange measurements were collected and analyzed from the genotypes, at 0, 30, and 60 days after the start of drought treatment.

Then, photosynthetic pigments were extracted using 100% pure acetone and filtered using filter paper and using a double beam, the optical density was measured using a UV–Vis spectrophotometer (Model 3092, Maharashtra, India) at 661.6 nm, 644.8 nm, and 470 nm. The contents of chlorophyll a (Chl-a), chlorophyll b (Chl-b), and total chlorophyll were computed following the calculation:

$$Chla = 12.25 A_{663.2} - 2.79 A_{646.8} \tag{19}$$

$$Chlb = 121.5 A_{646.8} - 5.10 A_{663.2} \tag{20}$$

$$Tchl = Chla + Chlb \tag{21}$$

where *Chl-a* is the content of chlorophyll-a (mg g$^{-1}$ tissue), *Chl-b* is the content chlorophyll-b (mg g$^{-1}$ tissue), and *Tchl* is total chlorophyll content (mg g$^{-1}$ tissue).

### 2.6. Cell Membrane Stability

Based on the works of Nijabat et al. [57], the cell membrane stability (CMS) of young leaves was determined at the end of drought stress experiments (plants at around 210 days of age) through relative conductivity. Fully expanded leaves were cut into 1 cm$^2$ pieces, washed with tap water and distilled water, then placed in a vial containing 10 mL de-ionized water for 18 h at 10 °C. Then, leaf discs were placed in vials at 25 °C, and in a water bath at 50 °C for 1 h 15 min. Then, the leaves were incubated at 15 °C for 18 h to facilitate the diffusion of electrolytes from leaf tissue to aqueous media. Then, the vials were brought to room temperature and initial conductance ($ws_1$ and $ww_1$) was measured after a brief shaking of the vials. Samples were then autoclaved at 0.10 MPa at 120 °C for 10 min, cooled down to 20 °C, contents were shaken, and final conductance ($ws_2$ and $ww_2$) was measured using a conductivity meter (HORIBA, model B-173, Kyoto, Japan).

Cell membrane stability (CMS, %) and relative cell injury (RCI, %) were calculated with the formulas:

$$CMS = \left( \frac{1 - \dfrac{ws1}{ws2}}{1 - \dfrac{ww1}{ww2}} \right) \times 100 \tag{22}$$

$$RCI = 100 - CMS \tag{23}$$

where *ws* and *ww* refer to conductance values for drought-stressed and well-watered coffee plants, respectively; and the numbers 1 and 2 refer to the initial and final conductance measurements, respectively.

### 2.7. Statistical Analysis

The collected data were subjected to statistical analysis using an analysis of variance test. Post hoc multiple comparisons were performed using Tukey's honest significant difference test ($p < 0.05$) to identify significant differences among the experimental groups. Pearson correlation analysis was performed after checking the assumptions of normality using the Shapiro–Wilk test. Principal component analyses were performed on all datasets using RStudio (version 4.2.1). All statistical analyses were performed using SigmaPlot version 13 (Systat Software Inc., San Jose, CA, USA). Moreover, Euclidean and Manhattan similarity index analyses were performed by Past 4.03 [58].

## 3. Results

### 3.1. Assessing C. arabica Seed Quality Traits of the Different Genotypes

The analyses of seeds' parameters possibly associated with germination potential (Figure 1, Table S2) showed significant differences ($p < 0.05$) in the seed's dry weight (Dw), seed length (Sl), seed surface area (SA), seed volume (SV), initial seed moisture content (Mc), and surface area to volume (SA to SV) ratio between the tested coffee genotypes, except for seed fresh weight (Fw) and seed width (Sw). The highest values for Dw (Figure 1A), Sl (Figure 1B), SA (Figure 1C), SV (Figure 1D), and Mc (Figure 1E) were recorded in the relatively tolerant genotype *Ca*74140 (0.174 g, 9.9 mm, 68.31 mm$^2$, 0.247 cm$^3$, and 14.89%, respectively), while the lowest values of those parameters were recorded in the sensitive genotype *Ca*754 (0.149 g, 7 mm, 51.8 mm$^2$, 0.201 cm$^3$, and 8.05%, respectively). Additionally, the highest SA to SV ratio (Figure 1F) of seeds was recorded in genotype *Ca*74140 (0.277), and the lowest in genotype *Ca*754 (0.258).

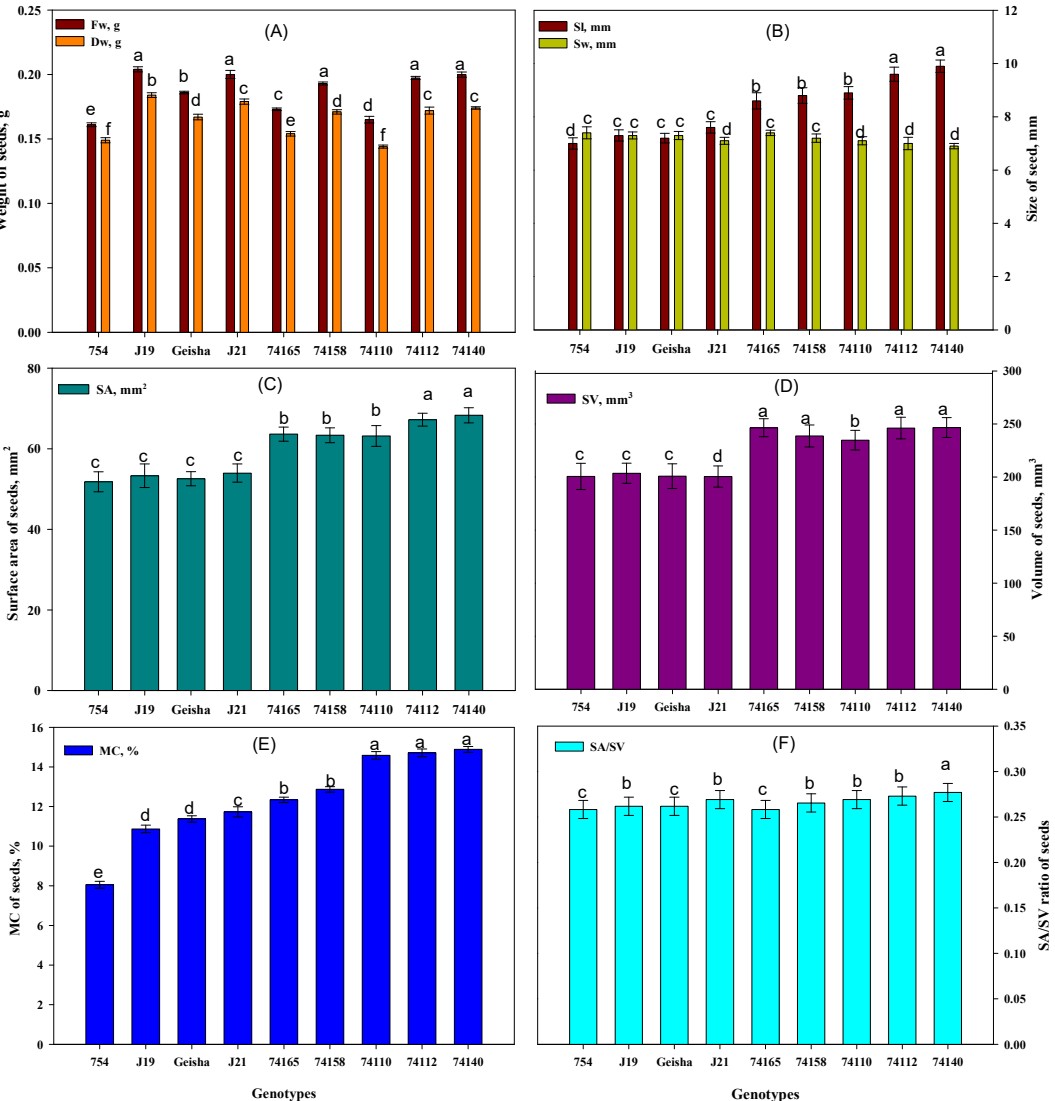

**Figure 1.** Pre-germination parameters of the seeds of nine *C. arabica* genotypes: (**A**) mean fresh (Fw) and dry (Dw) weight, (**B**) length (Sl) and width (Sw), (**C**) surface area (SA), (**D**) volume (SV), (**E**) moisture content (Mc), and (**F**) surface area to volume ratio (SA/SV). Bars indicate means ± SD, and the mean data are measurements of 60 representatives. Bars with the same letter indicates no significant difference at *p* < 0.05 between samples.

### 3.2. Assessment of Variabilities in Germination and Post-Germination Events of C. arabica

The comparison of the duration of germination and post-germination events of tested *C. arabica* genotypes revealed significant differences (Table 1). Genotype *Ca*74112, belonging to the relatively tolerant genotypes, had the shortest time to complete each germination stage, i.e., before germination stage-1 (*bg*-1) 3.2 days, before germination stage-2 (*bg*-2) 5.13 days, germination stage (*g*) 9.5 days, seedling development stage-1 (*sd*-1) 12.6 days, seedling development stage-2 (*sd*-2) 15.49 days, seedling development stage-3 (*sd*-3) 17.3 days, seedling development stage-4 (*sd*-4) 22.3 days, and seedling development stage-5 (*sd*-5) 44.26 days compared to the moderately sensitive and sensitive genotypes. The longest time to complete each germination stage was observed for the sensitive genotype *Ca*754 (*bg*-1 5.94 days, *bg*-2 11.1 days, *g* 17.52 days, *sd*-1 20 days, *sd*-2 23 days, *sd*-3 26 days, and *sd*-4 32 days), aside from the last stage, *sd*-5, which was the longest for *Ca*J19 (53.2 days). The other two relatively tolerant genotypes, i.e., *Ca*74110 and *Ca*74140, also had shorter periods to complete each developmental stage compared to the sensitive genotypes (Table 1).

**Table 1.** Chronological stages of before, during, and post-germination events of coffee seeds for nine *C. arabica* genotypes. Numbers represent means $\pm$ SD for *n* = 60 replicates per genotype. Numbers with the same superscript indicate no significant difference at *p*< 0.05 between samples.

| Stage Index | Stage Name | Average Period (Days) of Coffee Seed Developmental Stages | | | | | | | | |
|---|---|---|---|---|---|---|---|---|---|---|
| | | *Ca*754 | *Ca*J19 | *Ca*Geisha | *Ca*J21 | *Ca*74165 | *Ca*74158 | *Ca*74110 | *Ca*74112 | *Ca*74140 |
| *bg*-1 | Imbibition 1—primary imbibed seed (*bg*-1) | 5.94 $\pm$ 0.83 [b] | 5.70 $\pm$ 0.61 [b] | 6.01 $\pm$ 0.47 [c] | 5.3 $\pm$ 0.21 [b] | 5.2 $\pm$ 0.28 [b] | 5.3 $\pm$ 0.27 [b] | 4.01 $\pm$ 0.22 [a] | 3.2 $\pm$ 0.28 [a] | 4.2 $\pm$ 0.23 [a] |
| *bg*-2 | Imbibition 2— visible protuberance (*bg*-2) | 11.1 $\pm$ 0.71 [d] | 11.02 $\pm$ 0.63 [d] | 10.0 $\pm$ 0.52 [c] | 8.2 $\pm$ 0.37 [c] | 7.25 $\pm$ 0.32 [b] | 9.3 $\pm$ 0.44 [c] | 7.0 $\pm$ 0.29 [b] | 5.13 $\pm$ 0.47 [a] | 7.0 $\pm$ 0.34 [b] |
| *g* | Germinated seed (*g*) | 17.52 $\pm$ 0.27 [c] | 16.27 $\pm$ 0.21 [c] | 16.09 $\pm$ 0.36 [c] | 13.78 $\pm$ 0.32 [b] | 13.33 $\pm$ 0.25 [b] | 14.15 $\pm$ 0.35 [b] | 10.07 $\pm$ 0.26 [a] | 9.5 $\pm$ 0.24 [a] | 11.61 $\pm$ 0.24 [a] |
| *sd*-1 | Seedling 1—arrow-shaped radicle (*sd*-1) | 20 $\pm$ 2.16 [c] | 19.3 $\pm$ 2.86 [c] | 19.1 $\pm$ 2.51 [c] | 16 $\pm$ 1.98 [b] | 16 $\pm$ 1.64 [b] | 17.2 $\pm$ 1.87 [b] | 13.5 $\pm$ 1.92 [a] | 12.6 $\pm$ 1.98 [a] | 14.1 $\pm$ 1.61 [a] |
| *sd*-2 | Seedling 2—root primordia emergence (*sd*-2) | 23.0 $\pm$ 2.92 [c] | 22.0 $\pm$ 1.78 [c] | 22.0 $\pm$ 2.82 [c] | 19.11 $\pm$ 2.31 [b] | 19.13 $\pm$ 1.35 [b] | 20.2 $\pm$ 2.04 [b] | 16.53 $\pm$ 2.42 [a] | 15.49 $\pm$ 1.26 [a] | 17.36 $\pm$ 1.24 [a] |
| *sd*-3 | Seedling 3—lateral roots emergence (*sd*-3) | 26.0 $\pm$ 2.31 [d] | 25.0 $\pm$ 1.98 [d] | 25.26 $\pm$ 1.78 [d] | 22.7 $\pm$ 2.34 [c] | 22.2 $\pm$ 2.74 [c] | 23.1 $\pm$ 1.56 [c] | 19.2 $\pm$ 1.36 [b] | 17.3 $\pm$ 1.21 [a] | 20.14 $\pm$ 2.04 [b] |
| *sd*-4 | Seedling 4—lateral roots development (*sd*-4) | 32.0 $\pm$ 2.09 [d] | 31.1 $\pm$ 2.11 [d] | 30.21 $\pm$ 2.05 [d] | 28.2 $\pm$ 1.28 [c] | 27.05 $\pm$ 1.37 [c] | 28.1 $\pm$ 2.27 [c] | 24.6 $\pm$ 2.42 [b] | 22.3 $\pm$ 2.39 [a] | 25.0 $\pm$ 2.75 [b] |
| *sd*-5 | Seedling 5—photosynthetic leaves appear (*sd*-5) | 51.2 $\pm$ 3.07 [c] | 53.2 $\pm$ 3.86 [c] | 50.02 $\pm$ 3.54 [c] | 47.31 $\pm$ 3.22 [b] | 49.27 $\pm$ 2.84 [b] | 49.3 $\pm$ 3.05 [b] | 46.0 $\pm$ 3.01 [a] | 44.26 $\pm$ 3.21 [a] | 46.0 $\pm$ 2.23 [a] |

Note: The abbreviations bg, g, and sd stand for physiological and morphological changes before germination (bg-1 and bg-2), during germination (g), and seedling development (sd-1, sd-2, sd-3, sd-4, and sd-5), respectively.

Apart from the time needed for each coffee genotype to complete germination and post-germination stage, all the assessed genotypes displayed similar patterns of morphological changes (Figures 2 and 3, showing relatively tolerant *Ca*74112 as model genotype). At the *bg*-1 stage, the seeds of all genotypes fully imbibe water molecules without the appearance of a visible protuberance (Figure 2A). At the *bg*-2 stages, the visible protuberance with bulged root apex was noticed inside the endosperm cap caused by the elongation and growth of the embryo (Figure 2B). Next, due to the increased metabolic activities, growth of the embryo is highly facilitated, and the radicle with distinct features of the embryonic axis and remnants of suspensor emerges out of the endosperm, indicating the end of the germination phase (Figure 2C,D).

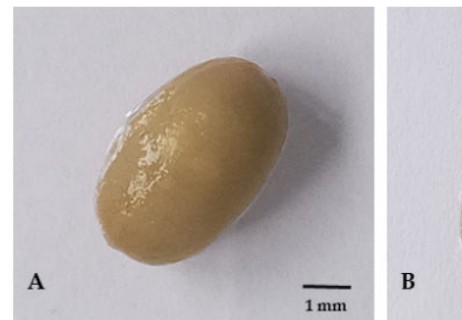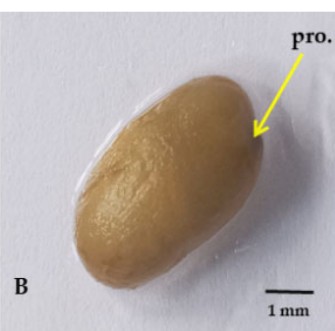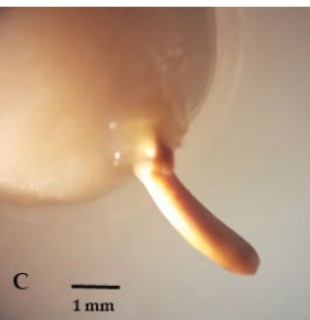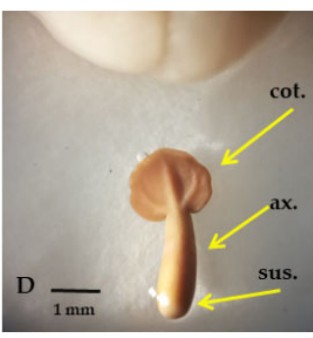

**Figure 2.** Photographs and micrographs of *C. arabica* germinating seed and embryo of relatively tolerant *Ca*74112 genotype: (**A**) seed during imbibition (*bg*-1), (**B**) imbibed seed with visible protuberance (pro.) (*bg*-2), (**C**) emergence of radicle from the outer layer of the endosperm (*g*), and (**D**) embryo with the cotyledons (cot.), the embryonic axis (ax.), and remnants of the suspensor (sus.) at the radicle tip (approximately 2–3.5 mm). Photos of (**A**,**B**) were taken using SonyAlphaA7RIV, and for (**C**,**D**), the observations were conducted under a Leica MZ8 microscope with a resolution power of 100 dpi.

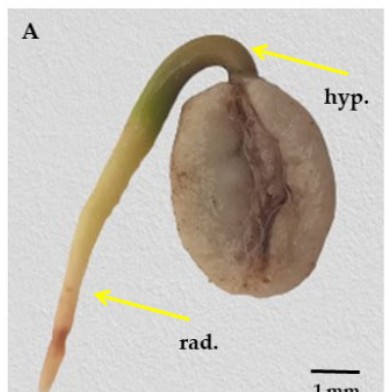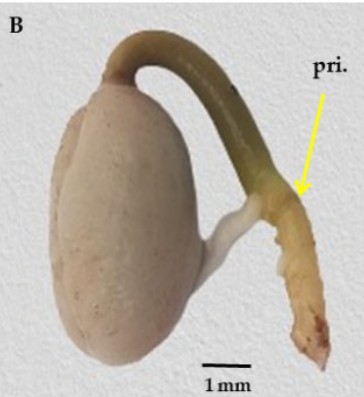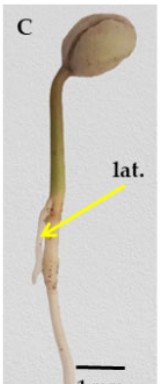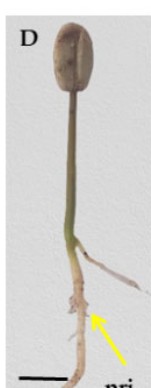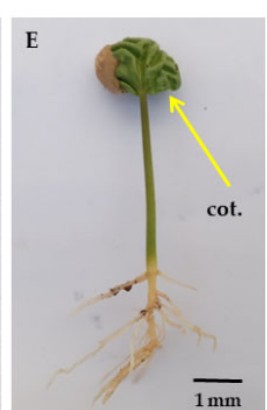

**Figure 3.** Photographs of post-germination stages of *C. arabica* development in chronological order of relatively tolerant *Ca*74112 genotype: (**A**) radicle (rad.) and hypocotyls (hyp.) emergence (*sd*-1), (**B**) root primordia (pri.) development between primary root and hypocotyls (*sd*-2), (**C**) lateral roots development (lat.) and appearance of root hairs on the primary root (*sd*-3), (**D**) properly developed primary (pri.) and lateral roots (*sd*-4), and (**E**) opening of cotyledonary (cot.) leaves (*sd*-5). Pictures were taken using SonyAlphaA7RIV.

At the sd-1 stage (Figure 3A), the endosperm area opposite the endosperm capis positioned towards gravity (positive geotropism) and begins to swell due to the growing cotyledonary leaves. Anthocyanine-driven, pink-colored hypocotyl begins to grow from the white arrow-shaped radicle, lifting the whole endosperm upward from the germinating media indicating an epigeal type of germination. At the sd-2 stage (Figure 3B), root primordia appeared at the junction between the hypocotyl and primary root, and the radicle and hypocotyl becomes enlarged. At the sd-3 stage (Figure 3C), the furrow in the

outer endosperm begins to split and crack due to the growing pressure of the cotyledonary leaves. The endosperm begins to erect, the hypocotyl begins to change color into green, lateral roots develop from the root primordial regions, and additional root primordia and root hairs develop from the primary root. At the sd-4 stage (Figure 3D), the outer structural parts of the endosperm become soft, flaccid, and loose. The structure of the endosperm containing the cotyledonary leaves becomes erect (positive phototropism) and hypocotyl begins changing color into green. Moreover, properly developed primary and lateral roots are present, and several root hairs appeared on the surface of the root. At the sd-5 stage (Figure 3E), the endosperm begins to disappear, folded cotyledonary leaves begin to open, and an increased number and size of primary and lateral roots are observed.

### 3.3. Assessing Seed Germination Potential Indicators of the Different Genotypes

Significant variations in the germination parameters were recorded among the nine *C. arabica* genotypes analyzed in this study. In general, relatively tolerant genotypes *Ca*74140, *Ca*74112, and *Ca*74110 demonstrated higher germination performances compared to moderately sensitive (*Ca*74158, *Ca*74165, and *Ca*J-21), and sensitive (*Ca*754, *Ca*J-19, and *Ca*Geisha) genotypes (Figure 4, Table S3).

Germination percentage (GP) is an estimate of the germinability of the population of seeds [40], and the GP values calculated for tested coffee genotypes were as follows: relatively tolerant, i.e., *Ca*74140 (90 ± 1.44%), *Ca*74112 (80 ± 1.74%), and *Ca*74110 (75 ± 1.56%), moderately sensitive, i.e., *Ca*74158 (65 ± 1.85%), *Ca*74165 (60 ± 1.75%), and *Ca*J-21 (55 ± 2.92%), and sensitive, i.e., *Ca*Geisha (50 ± 2.46%), *Ca*J-19 (45 ± 2.21%), and *Ca*754 (35 ± 2.67%) (Figure 4A). Mean germination time (MGT), which is a measure of the time it takes for the seed to germinate focusing on the day by which most seeds have germinated [41], was the shortest for the relatively tolerant genotype *Ca*74112 (9.50 days) and significantly longer for the sensitive genotype *Ca*754 (17.52 days) (Figure 4B). CVG focuses on the time required to reach the final germination percentage [44], and $CV_t$ interprets and calculates the coefficient of variation of the mean germination time [43]. Hence, in terms of CVG and $CV_t$, the highest values were recorded in the relatively tolerant genotype *Ca*74112 (10.53% and 20.94%, respectively) and the lowest values were observed in sensitive genotype *Ca*754 (5.71% and 2.92%, respectively) (Figure 4C). GRI describes the percentage of germination per day, the higher the percentage and the shorter the duration, the higher the GRI [46], whereas germination index (GI) is an estimate of the time (in days) that it takes a certain germination percentage to occur [45]. In this study, the highest and lowest value of GRI and GI were recorded in the relatively tolerant genotype *Ca*74112 (8.75%/day and 5.25 seed/day, respectively) and the lowest in the sensitive genotype *Ca*754 (1.99%/day and 1.2 seed/day, respectively) (Figure 4D).

Uncertainty of germination (U) indicates the degree of uncertainty associated with the distribution of relative frequency of germination [48], and synchrony of germination process (Z) describes the degree of overlapping of germination among seeds [49]. The highest U values were recorded in the relatively tolerant genotype *Ca*74110/*Ca*74112 (2.61 bit), whereas the lowest value was observed for the moderately sensitive genotype *Ca*J-21 (0.76 bit) (Figure 4E). The highest Z value was recorded in the moderately sensitive genotype *Ca*J-21 (0.64) and the lowest for the relatively tolerant genotype *Ca*74112 (0.19) (Figure 4E). The mean daily germination (MDG) percent represents the mean number of seeds germinated per day [50]. The MDG of the relatively tolerant genotype *Ca*74140 exhibited the highest (3.60%) and sensitive genotype *Ca*754 showed the lowest value (1.40%) (Figure 4F). The peak value (Pv) of germination is the accumulated number of seeds germinated at the point on the germination curve at which the rate of germination starts to decrease [50], and germination value (Gv) is the combination of speed and completeness of germination [51]. The relatively tolerant genotype *Ca*74140 exhibited the highest Pv and Gv (6.54% day$^{-1}$ and 23.54%$^2$ day$^{-1}$, respectively) and sensitive genotype *Ca*754 showed the lowest Pv and Gv values (1.94% day$^{-1}$ and 2.72%$^2$ day$^{-1}$, respectively) (Figure 4G).

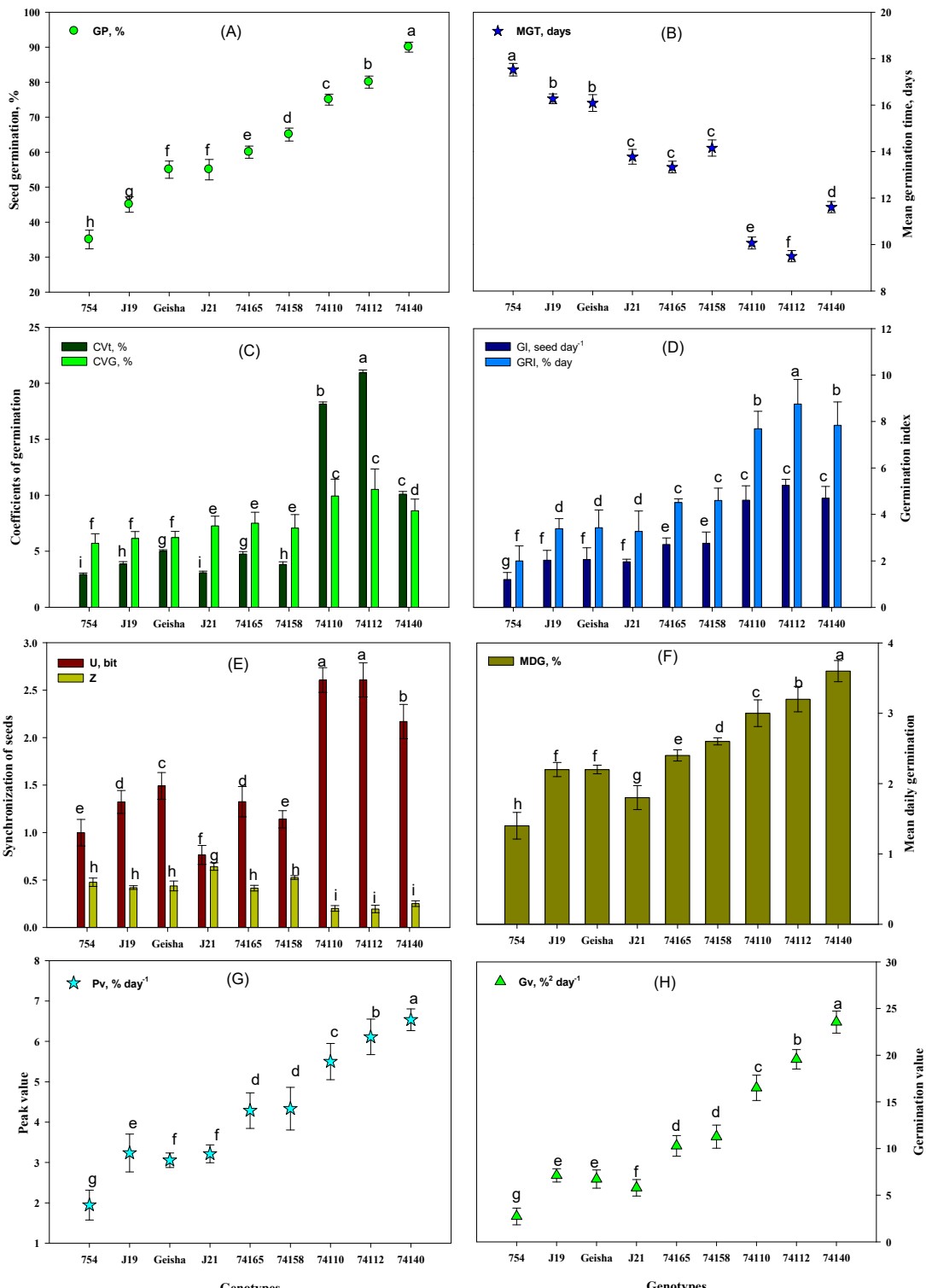

**Figure 4.** Germination parameters of the nine *C. arabica* genotypes: (**A**) mean germination percentage (GP), (**B**) mean germination time (MGT), (**C**) coefficient of variation of germination time ($CV_t$) and coefficient of the velocity of germination (CVG), (**D**) germination index (GI) and germination rate index (GRI), (**E**) uncertainty of germination process (U) and synchronization index (Z), (**F**) mean daily germination percent (MDG), (**G**) peak value for germination (Pv), and (**H**) germination value (Gv). Dots and bars indicate means ± SD (*n* = 60 replicates per genotype). Dots and bars with the same letter indicate no significant different at *p* < 0.05 between samples.

*3.4. Growth and Development of C. arabica Seedlings*

Coffee seedlings were collected after 90 days of germination. The result revealed significant differences in seedlings of different genotypes as shown by root and shoot length (Rl and SdL), root number (RN), and vigorous index (VI) (Table 2). The highest Rl, SdL, and Rn were recorded in the relatively tolerant genotypes *Ca*74112 (67.5 mm, 50 mm, and 22.5, respectively), *Ca*74110 (65.75 mm, 49.75 mm, and 22.25, respectively), and *Ca*74140 (62.75 mm, 48 mm, and 22.75, respectively), while the lowest was observed in sensitive genotype *Ca*754 (45 mm, 42.75 mm, and 13.5, respectively). Additionally, the highest root-to-shoot ratios (R/S$^r$) were calculated in relatively tolerant genotypes *Ca*74112 (1.35), *Ca*74110 (1.32), and *Ca*74140 (1.31), and the lowest was found in sensitive genotype *Ca*754 (1.05). Moreover, relatively tolerant genotype *Ca*74140 had the highest VI (9967.5), followed by *Ca*74112 (9400) and *Ca*74110 (8662.5), while the significantly lower VI was recorded for the sensitive genotype *Ca*754 (3071.25) (Table 2).

**Table 2.** Root length, shoot length, root number, root–shoot ratio, and vigorous index of coffee seedlings 90 days after germination. The numbers are means ± SD of 60 representatives of each genotype. Numbers with the same superscript indicate no significant difference at *p* < 0.05 between samples.

| Genotypes | RL (mm) | SdL (mm) | RN | R/S$^r$ | VI |
|---|---|---|---|---|---|
| *Ca*754 | 45 ± 0.41 [c] | 42.75 ± 0.1 [b] | 13.5 ± 1.19 [c] | 1.05 [d] | 3071.25 [d] |
| *Ca*J19 | 51.75 ± 0.21 [b] | 42.75 ± 0.25 [c] | 14 ± 2.46 [b] | 1.21 [c] | 4252.5 [c] |
| *Ca*Geisha | 52.5 ± 0.51 [c] | 43.5 ± 0.23 [b] | 15 ± 1.47 [c] | 1.21 [c] | 5280 [c] |
| *Ca*J21 | 59 ± 0.21 [c] | 46.5 ± 0.25 [b] | 19.25 ± 2.46 [c] | 1.27 [b] | 5802.5 [c] |
| *Ca*74165 | 60.75 ± 0.41 [b] | 47.75 ± 0.23 [c] | 21.75 ± 2.75 [b] | 1.27 [b] | 6510 [b] |
| *Ca*74158 | 53.75 ± 0.53 [c] | 44.25 ± 0.21 [b] | 16 ± 1.58 [b] | 1.21 [b] | 6370 [b] |
| *Ca*74110 | 65.75 ± 0.31 [a] | 49.75 ± 0.09 [a] | 22.25 ± 2.66 [a] | 1.32 [a] | 8662.5 [a] |
| *Ca*74112 | 67.5 ± 0.30 [a] | 50 ± 0.21 [a] | 22.5 ± 3.08 [a] | 1.35 [a] | 9400 [a] |
| *Ca*74140 | 62.75 ± 0.90 [a] | 48 ± 0.20 [a] | 22.75 ± 1.75 [a] | 1.31 [a] | 9967.5 [a] |

Note: RL—root length, SdL—seedling shoot length, RN—root number, R/S$^r$—root-to-shoot ratio, and VI—vigorous index.

*3.5. Assessing the Effect of Drought Stress on the Growth and Physiology of Adult Coffee Genotypes*

3.5.1. Shoot Growth of Coffee Plants in Control and Drought Stress Conditions

Compared with the well-water plants, significant reductions were recorded in stem length, stem diameter, leaf number, and leaf area among the genotypes growing under drought stress conditions. Under drought stress conditions, the highest stem length, stem diameter, leaf number, and leaf area (Table 3) were recorded in the relatively tolerant genotype *Ca*74112 (18.14 ± 0.04 cm, 3.34 ± 0.08 cm, 9.5 ± 0.5, and 18.49 ± 0.38 cm$^2$), and the lowest values of stem length, leaf number, and leaf area were recorded for sensitive genotypes *Ca*754 (11.38 ± 0.3 cm, 8, 10.72 ± 0.18 cm$^2$) and stem diameter for *Ca*J-19 (2.84 ± 0.08 cm). Drought stress-induced minimum and maximum stem elongation and leaf area expansion were recorded in the genotype *Ca*754 (49.16%, 53.99%) and *Ca*74112 (61.67%, 68.81%), respectively (Figure S6; Tables S4–S7 and S18).

**Table 3.** The mean stem height, stem diameter, leaf number, and leaf area of plants belonging to the four coffee genotypes, *Ca754*, *Ca*J-19, *Ca74110*, and *Ca74112*, under well-water (*ww*) and drought stress conditions (*ws*), after 60 days of the study. The numbers are means $\pm$ SD ($n$ = 15 replicates per genotype). Numbers with the same superscript indicate no significant difference at $p < 0.05$ between samples.

| Variable | Time (DADB) | Ca754 | | CaJ-19 | | Ca74110 | | Ca74112 | |
|---|---|---|---|---|---|---|---|---|---|
| | | *ww* | *ws* | *ww* | *ws* | *ww* | *ws* | *ww* | *ws* |
| Stem height (cm) | 0 | 3 ± 0.20 [a] | 3 ± 0.24 [a] | 4.9 ± 0.09 [b] | 4.88 ± 0.21 [b] | 7.1 ± 0.2 [c] | 7 ± 0.13 [c] | 9.28 ± 0.08 [d] | 9.75 ± 0.05 [d] |
| | 10 | 4.25 ± 0.13 [a] | 4.2 ± 0.31 [a] | 6.13 ± 0.05 [b] | 6.12 ± 0.2 [b] | 8.3 ± 0.21 [c] | 8.22 ± 0.11 [c] | 10.49 ± 0.08 [d] | 10.96 ± 0.06 [d] |
| | 20 | 7.68 ± 0.14 [a] | 7.57 ± 0.28 [a] | 9.53 ± 0.06 [b] | 9.45 ± 0.23 [b] | 11.7 ± 0.2 [c] | 11.54 ± 0.08 [c] | 13.85 ± 0.09 [d] | 14.32 ± 0.04 [d] |
| | 30 | 11.61 ± 0.15 [a] | 8.59 ± 0.28 [e] | 13.48 ± 0.05 [b] | 10.45 ± 0.2 [a] | 15.65 ± 0.18 [c] | 12.55 ± 0.11 [b] | 17.8 ± 0.06 [d] | 15.31 ± 0.04 [c] |
| | 40 | 15.7 ± 0.15 [a] | 9.88 ± 0.26 [c] | 17.58 ± 0.07 [a] | 11.75 ± 0.21 [c] | 19.76 ± 0.2 [b] | 13.85 ± 0.11 [d] | 21.92 ± 0.08 [b] | 16.6 ± 0.03 [e] |
| | 50 | 18.08 ± 0.17 [a] | 10.91 ± 0.29 [c] | 19.99 ± 0.07 [a] | 12.76 ± 0.21 [c] | 22.15 ± 0.2 [b] | 14.86 ± 0.12 [d] | 24.3 ± 0.06 [b] | 17.6 ± 0.04 [e] |
| | 60 | 23.15 ± 0.20 [a] | 11.38 ± 0.3 [c] | 25.08 ± 0.09 [a] | 13.25 ± 0.19 [c] | 27.21 ± 0.18 [b] | 15.35 ± 0.09 [d] | 29.41 ± 0.07 [b] | 18.14 ± 0.04 [e] |
| Stem diameter (mm) | 0 | 1.96 ± 0.05 [a] | 1.9 ± 0.11 [a] | 1.57 ± 0.06 [a] | 1.61 ± 0.08 [a] | 2.05 ± 0.09 [a] | 2.04 ± 0.21 [a] | 2.23 ± 0.16 [a] | 2.08 ± 0.05 [a] |
| | 10 | 2.43 ± 0.03 [a] | 2.41 ± 0.13 [a] | 1.98 ± 0.09 [a] | 2.03 ± 0.12 [a] | 2.58 ± 0.08 [b] | 2.52 ± 0.23 [b] | 2.71 ± 0.19 [b] | 2.56 ± 0.05 [b] |
| | 20 | 2.9 ± 0.06 [a] | 2.8 ± 0.11 [a] | 2.48 ± 0.06 [b] | 2.53 ± 0.09 [b] | 3 ± 0.09 [c] | 2.97 ± 0.2 [c] | 3.18 ± 0.2 [c] | 2.9 ± 0.09 [c] |
| | 30 | 3.44 ± 0.05 [c] | 3.03 ± 0.12 [a] | 3.0 ± 0.07 [a] | 2.74 ± 0.07 [a] | 3.51 ± 0.05 [d] | 3.17 ± 0.22 [a] | 3.69 ± 0.19 [e] | 3.26 ± 0.07 [b] |
| | 40 | 3.88 ± 0.05 [a] | 3.07 ± 0.11 [c] | 3.47 ± 0.06 [a] | 2.76 ± 0.08 [c] | 3.98 ± 0.09 [b] | 3.20 ± 0.22 [d] | 4.16 ± 0.2 [b] | 3.29 ± 0.07 [d] |
| | 50 | 4.39 ± 0.03 [a] | 3.12 ± 0.11 [c] | 3.98 ± 0.07 [a] | 2.82 ± 0.09 [c] | 4.49 ± 0.05 [b] | 3.24 ± 0.22 [d] | 4.69 ± 0.18 [b] | 3.31 ± 0.07 [d] |
| | 60 | 4.88 ± 0.05 [a] | 3.11 ± 0.12 [b] | 4.49 ± 0.06 [a] | 2.84 ± 0.08 [b] | 5.02 ± 0.05 [a] | 3.27 ± 0.21 [c] | 5.19 ± 0.18 [a] | 3.34 ± 0.08 [d] |
| Leaf number | 0 | 6 ± 0.0 [a] | 6 ± 0.0 [a] | 6.5 ± 0.5 [a] | 6.5 ± 0.4 [a] | 7 ± 0.51 [b] | 7 ± 0.54 [b] | 7.5 ± 0.5 [c] | 7.5 ± 0.3 [c] |
| | 10 | 8 ± 0.0 [a] | 8 ± 0.0 [a] | 8.5 ± 0.4 [b] | 8.5 ± 0.5 [b] | 9 ± 0.53 [c] | 9 ± 0.58 [c] | 9.5 ± 0.5 [d] | 9.5 ± 0.3 [d] |
| | 20 | 8 ± 0.0 [a] | 8 ± 0.0 [a] | 8.5 ± 0.5 [b] | 8.5 ± 0.3 [b] | 9 ± 0.58 [c] | 9 ± 0.58 [c] | 9.5 ± 0.6 [d] | 9.5 ± 0.7 [d] |
| | 30 | 8 ± 0.0 [a] | 8 ± 0.0 [a] | 8.5 ± 0.3 [b] | 8.5 ± 0.5 [b] | 9 ± 0.55 [c] | 9 ± 0.55 [c] | 9.5 ± 0.5 [d] | 9.5 ± 0.5 [d] |
| | 40 | 10 ± 0.0 [a] | 8 ± 0.0 [c] | 10.5 ± 0.5 [a] | 8.5 ± 0.4 [c] | 11 ± 0.56 [b] | 9 ± 0.56 [d] | 11.5 ± 0.4 [b] | 9.5 ± 0.5 [d] |
| | 50 | 12 ± 0.0 [a] | 8 ± 0.0 [c] | 12.5 ± 0.6 [a] | 8.5 ± 0.5 [c] | 13 ± 0.50 [b] | 9 ± 0.56 [d] | 13.5 ± 0.5 [b] | 9.5 ± 0.4 [d] |
| | 60 | 16 ± 0.0 [a] | 8 ± 0.0 [c] | 16.5 ± 0.2 [a] | 8.5 ± 0.4 [c] | 17 ± 0.58 [b] | 9 ± 0.52 [d] | 17.5 ± 0.7 [b] | 9.5 ± 0.4 [e] |
| Leaf area (cm$^2$) | 0 | 7.75 ± 0.73 [a] | 7.68 ± 0.18 [a] | 8.35 ± 1.53 [a] | 8.24 ± 0.68 [a] | 11.16 ± 1.63 [b] | 10.83 ± 0.72 [b] | 14.79 ± 0.75 [c] | 15.42 ± 0.38 [c] |
| | 10 | 9.14 ± 0.74 [a] | 9.01 ± 0.18 [a] | 9.74 ± 1.56 [b] | 9.63 ± 0.71 [b] | 12.52 ± 1.64 [c] | 12.16 ± 0.73 [c] | 16.16 ± 0.75 [d] | 16.75 ± 0.38 [d] |
| | 20 | 10.73 ± 0.73 [a] | 10.31 ± 0.2 [a] | 11.33 ± 1.53 [b] | 10.95 ± 0.74 [b] | 14.11 ± 1.62 [c] | 13.48 ± 0.73 [c] | 17.75 ± 0.77 [d] | 18.05 ± 0.4 [d] |
| | 30 | 12.59 ± 0.73 [a] | 10.58 ± 0.18 [b] | 13.19 ± 1.53 [a] | 11.22 ± 0.71 [b] | 15.97 ± 1.61 [c] | 13.75 ± 0.73 [a] | 19.66 ± 0.74 [d] | 18.36 ± 0.39 [d] |
| | 40 | 14.51 ± 0.73 [a] | 10.67 ± 0.18 [d] | 15.11 ± 1.53 [a] | 11.32 ± 0.72 [d] | 17.89 ± 1.62 [b] | 13.86 ± 0.72 [a] | 21.53 ± 0.76 [c] | 18.45 ± 0.38 [e] |
| | 50 | 17.07 ± 0.74 [a] | 10.7 ± 0.18 [c] | 17.68 ± 1.53 [a] | 11.34 ± 0.72 [c] | 20.45 ± 1.62 [b] | 13.88 ± 0.72 [d] | 24.08 ± 0.75 [b] | 18.47 ± 0.38 [e] |
| | 60 | 19.85 ± 0.7 [a] | 10.72 ± 0.18 [d] | 20.44 ± 1.57 [a] | 11.36 ± 0.72 [d] | 23.23 ± 1.62 [b] | 13.9 ± 0.72 [d] | 26.87 ± 0.77 [c] | 18.49 ± 0.38 [e] |

Note: *ww*—well-water; *ws*—drought stress.

### 3.5.2. Root Growth of Coffee Plants in Control and Drought Stress Conditions

Drought stress negatively affects root traits in terms of root length, root number, and root volume. Those parameters were significantly lower for plants growing under drought stress than those growing under well-watered conditions. The impact of drought stress on the tested genotypes was also significantly different, and the highest and lowest root length (Figure 5A), root number (Figure 5B), and root volume (Figure 5C) were recorded in the relatively tolerant genotype of *Ca*74112 (16.33 ± 1.53 cm, 40.15 ± 1.55, 5.87 ± 1.57 cm$^3$, respectively) and sensitive genotype of *Ca*754 (10.19 ± 0.21 cm, 26.51 ± 0.23, 3.15 ± 0.21 cm$^3$, respectively), respectively. As a result of drought stress, maximum and minimum increments of root length, number, and volume were identified in the relatively tolerant genotype of *Ca*74112 (77.15%, 78.11%, 78.25%, respectively) and *Ca*J-19 (57.26%, 56.05%, 58.36%, respectively), respectively (Figure S7; Table S8).

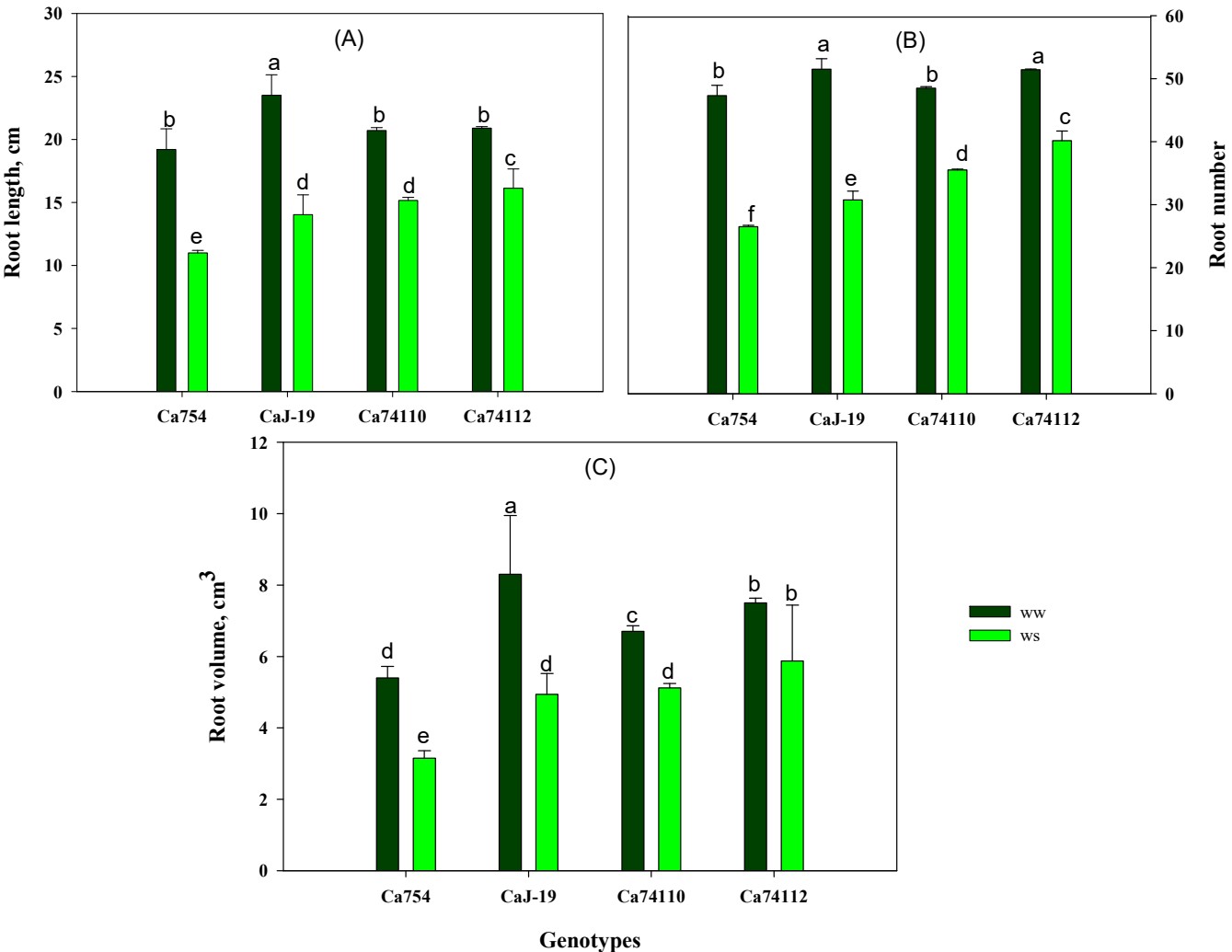

**Figure 5.** Root parameters of four *C. arabica* genotypes grown under well-water (*ww*) and drought stress (*ws*) conditions, after 60 days of drought treatment: (**A**) root length (RL), (**B**) root number (RN), and (**C**) root volume (RV). Bars indicate mean ± SD (*n* = 15 replicates per genotype). Bars with the same letter indicate no significant difference at *p* < 0.05 between samples.

### 3.5.3. Biomass of Coffee Plants in Control and Drought Stress Conditions

At the end of the study, significantly lower fresh and dry weights were recorded in all genotypes grown under drought stress than in well-watered conditions. Under drought stress conditions, the highest and lowest root, leaf, stem, and total fresh mass were measured in the relatively tolerant genotype *Ca*74112 (3.03 ± 0.48 g; 3.81 ± 0.75 g;

2.46 ± 1.12 g; 9.3 g, respectively) and sensitive genotype *Ca*754 (1.51 ± 0.11 g; 1.63 ± 0.26 g; 1.28 ± 0.13 g; 4.43 g, respectively), respectively. Similarly, the highest and lowest root, leaf, stem, and total dry mass were measured in the relatively tolerant genotype of *Ca*74112 (0.88 ± 0.03 g; 1.11 ± 0.04 g; 0.72 ± 0.01 g; 2.71 g, respectively) and sensitive genotypes of *Ca*754 (0.4 ± 0.03 g; 0.43 ± 0.01 g; 0.34 ± 0.06 g; 1.17 g, respectively), respectively (Table 4) (Figure S8).

**Table 4.** Mean variables of the fresh and dry weight of the four adult coffee genotypes, *Ca*754, *Ca*J-19, *Ca*74110, and *Ca*74112, under well-water (*ww*) and drought stress (*ws*) conditions, after 60 days of the study. The numbers are means ± SD (*n* = 15 replicates per genotype). Numbers with the same superscript indicate no significant difference at *p* < 0.05 between samples.

| Variable | *Ca*754 | | *Ca*J-19 | | *Ca*74110 | | *Ca*74112 | |
|---|---|---|---|---|---|---|---|---|
| | *ww* | *ws* | *ww* | *ws* | *ww* | *ws* | *ww* | *ws* |
| RFM, g | 4.12 ± 0.26 [b] | 1.51 ± 0.11 [e] | 4.65 ± 0.2 [a] | 1.85 ± 0.06 [e] | 4.15 ± 0.26 [b] | 2.21 ± 0.06 [d] | 4.23 ± 0.07 [a] | 3.03 ± 0.48 [c] |
| LFM, g | 4.38 ± 0.03 [b] | 1.63 ± 0.26 [e] | 5.45 ± 0.17 [a] | 2.13 ± 0.06 [d] | 4.67 ± 0.03 [b] | 2.75 ± 0.06 [d] | 5.25 ± 0.06 [a] | 3.81 ± 0.75 [c] |
| SFM, g | 3.2 ± 0.18 [b] | 1.28 ± 0.13 [d] | 4.3 ± 0.18 [a] | 1.76 ± 0.06 [d] | 3.3 ± 0.18 [b] | 2.16 ± 0.06 [c] | 4 ± 0.05 [a] | 2.46 ± 1.12 [c] |
| TFM, g | 11.7 ± 0.77 [b] | 4.43 ± 0.08 [f] | 14.4 ± 0.18 [a] | 5.73 ± 0.07 [e] | 12.12 ± 0.77 [b] | 7.11 ± 0.07 [d] | 13.48 ± 0.07 [a] | 9.3 ± 0.1 [c] |
| RDM, g | 1.09 ± 0.06 [a] | 0.4 ± 0.03 [c] | 1.28 ± 0.18 [a] | 0.51 ± 0.06 [b] | 1.17 ± 0.03 [a] | 0.62 ± 0.07 [b] | 1.23 ± 0.01 [a] | 0.88 ± 0.03 [b] |
| LDM, g | 1.16 ± 0.04 [b] | 0.43 ± 0.01 [d] | 1.49 ± 0.18 [a] | 0.58 ± 0.06 [c] | 1.32 ± 0.01 [b] | 0.78 ± 0.06 [c] | 1.53 ± 0.02 [a] | 1.11 ± 0.04 [b] |
| SDM, g | 0.84 ± 0.04 [b] | 0.34 ± 0.06 [d] | 1.18 ± 0.06 [a] | 0.48 ± 0.06 [d] | 0.93 ± 0.03 [b] | 0.61 ± 0.06 [c] | 1.16 ± 0.04 [a] | 0.72 ± 0.01 [c] |
| TDM, g | 3.09 ± 0.03 [a] | 1.17 ± 0.06 [c] | 3.95 ± 0.06 [a] | 1.57 ± 0.07 [c] | 3.42 ± 0.04 [a] | 2.01 ± 0.05 [b] | 3.92 ± 0.02 [a] | 2.71 ± 0.03 [b] |

Note: *ww*—well-water; *ws*—drought stress; RFM—root fresh mass; LFM—leaf fresh mass; SFM—stem fresh mass; TFM—total fresh mass; RDM—root dry mass; LDM—leaf dry mass; SDM—stem dry mass; and, TDM—total dry mass.

### 3.5.4. Relative Water Content and Stem Water Potential

Drought stress significantly (*p* < 0.05) lowered relative water content (RWC) and water potential ($\Psi_w$,–Mpa) (Figure 6). Under drought stress, the mean RWC was higher in the relatively tolerant genotypes of *Ca*74112 (48.09 ± 0.8%) and *Ca*74110 (43.4 ± 0.29%), and lower in the sensitive genotypes of *Ca*J-19 (32.57 ± 0.13%) and *Ca*754 (30.24 ± 0.21%) (Figure 6A). As a result of drought stress, the minimum and maximum reduction of RWC under drought stress were recorded in the relatively tolerant genotypes of *Ca*74112 (41.89%) and *Ca*74110 (46.74%) and sensitive genotypes of *Ca*J-19 (60.32%) and *Ca*754 (62.74%), respectively (Tables S9 and S18).

Throughout the duration of the experiment, the water potential of the genotypes in adequately irrigated plants exhibited no significant differences (ranging between −1.44 Mpa to −1.48 Mpa). In plants grown in drought stress conditions, prolonged exposure of plants to drought stress resulted in a significant decrease in the stem water potential. Under drought stress conditions, at the end of the experiment, the mean $\Psi_w$ was significantly (*p* < 0.001) lower in all genotypes, and the higher $\Psi_w$ was maintained in the relatively tolerant genotypes of *Ca*74112 (−2.56 ± 0.02, Mpa) and *Ca*74110 (−2.64 ± 0.02, Mpa), whereas in the sensitive genotypes, the decrease was more apparent, i.e., *Ca*J-19 (−3.02 ± 0.07, Mpa) and *Ca*754 (−3.11 ± 0.02, Mpa) (Figure 6B). Fluctuations in $\Psi_w$ revealed a contrasting behavior among the genotypes. As a result of drought stress, the percentage of the decrease in $\Psi_w$ among the tested genotypes from highest to lowest was *Ca*754 (53.24%), *Ca*J-19 (52.05%), *Ca*74110 (45.38%), and *Ca*74112 (42.14%) (Table S10).

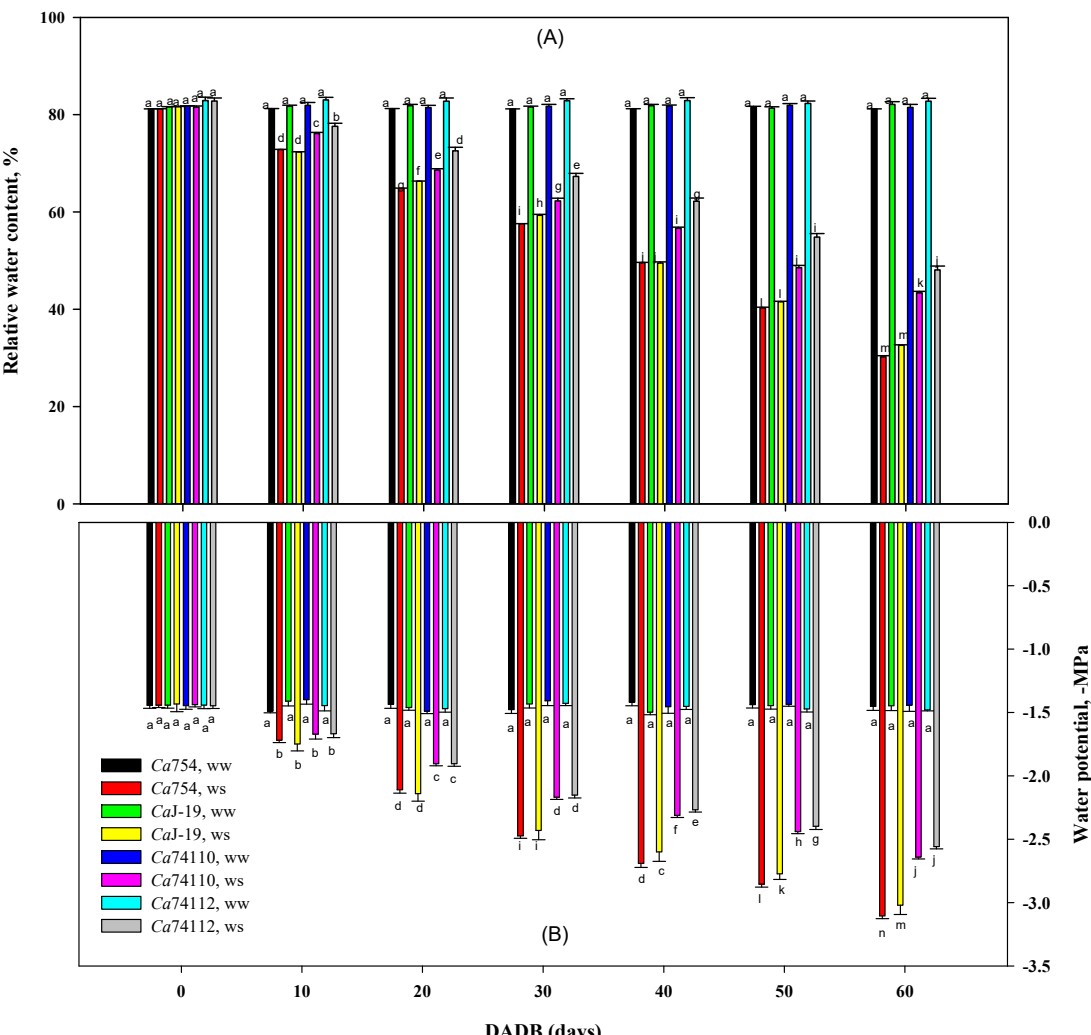

**Figure 6.** Effects of drought treatment on (**A**) relative water content (RWC), and (**B**) water potential of the four adult coffee genotypes, *Ca*754, *Ca*J-19, *Ca*74110, and *Ca*74112, under well-water (*ww*) and drought stress conditions (*ws*). Bars indicate means ± SD (*n* = 3 replicates per genotype). Bars with the same letter indicate no significant difference at *p* < 0.05 between samples. DADB indicates the number of days after drought stress begins.

### 3.5.5. Photosynthesis Assimilation Rate, Stomatal Conductance, and Transpiration Rate

In all genotypes, as the stress intensified, the $CO_2$ assimilation rate (A), transpiration rate (E), and stomatal conductance (Gs) were significantly decreased in plants grown under drought stress as compared to those growing under well-watered conditions (Figure 7).

In plants grown under control conditions, throughout the study period, the $CO_2$ assimilation rate was not significantly different among the genotypes, with a range of 7.24 to 7.59 µmol m$^{-2}$s$^{-1}$. Under drought stress conditions, the genotypes showed significantly different reductions in mean $CO_2$ assimilation rate. The higher A values were recorded in the relatively tolerant genotypes of *Ca*74112 (2.89 ± 0.11 µmol m$^{-2}$s$^{-1}$) and *Ca*74110 (2.29 ± 0.06 µmol m$^{-2}$s$^{-1}$), and the lower values in the sensitive genotypes *Ca*J-19 (1.55 ± 0.13 µmol m$^{-2}$s$^{-1}$) and *Ca*754 (1.00 ± 0.09 µmol m$^{-2}$s$^{-1}$) (Figure 7A). Comparing the reduction of $CO_2$ assimilation rate as a result of drought stress, significantly, the highest rate of reduction was recorded in the sensitive genotype *Ca*754 (86.85%), whereas the lowest was in tolerant genotype *Ca*74112 (61.96%) (Tables S11 and S18).

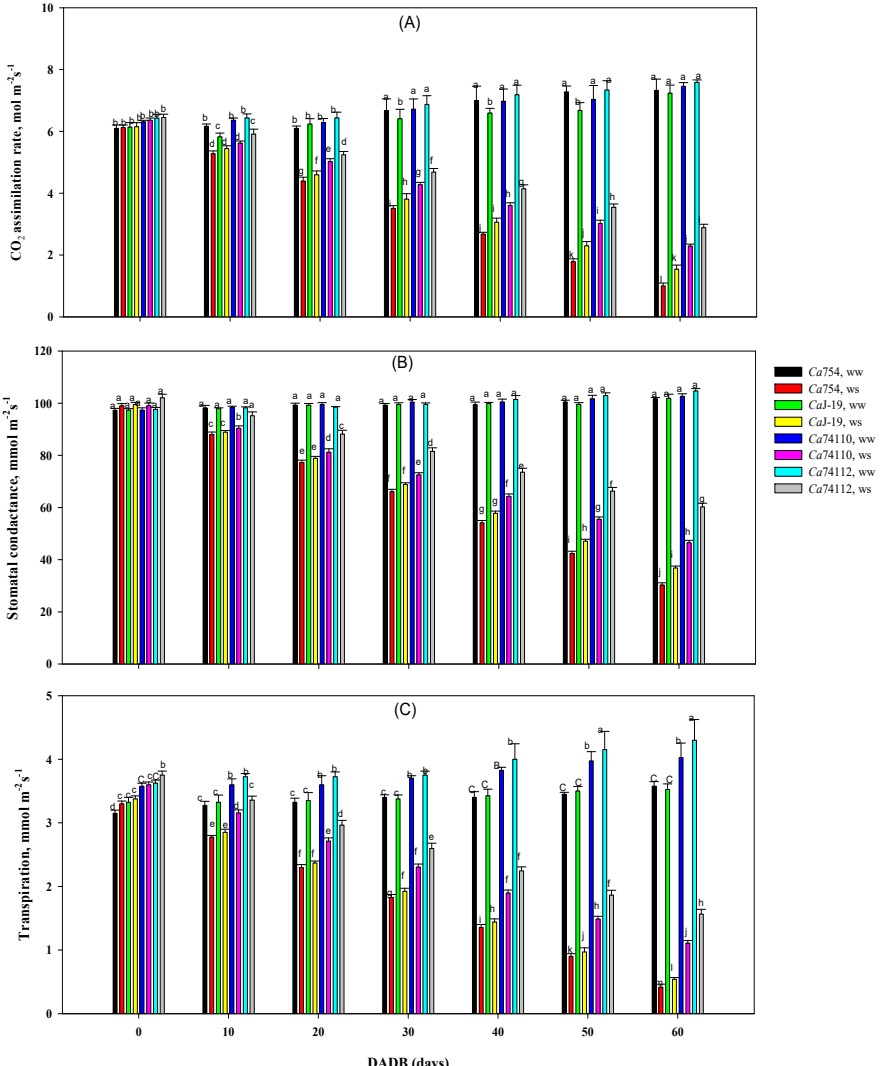

**Figure 7.** Effects of drought stress on (**A**) $CO_2$ assimilation rate, (**B**) stomatal conductance, and (**C**) transpiration rate of the four adult coffee genotypes, *Ca*754, *Ca*J-19, *Ca*74110, and *Ca*74112, under well-water (*ww*) and drought stress conditions (*ws*). Bars indicate means ± SD (*n* = 15 replicates per genotype). Bars with the same letter indicate no significant difference at *p* < 0.05 between samples. DADB indicates the number of days after drought stress begins.

There were no significant differences in the stomatal conductance values when comparing plants grown under drought stress and well-watered conditions at the early stage, but at the end of the experiment, significant differences in Gs were displayed by the genotypes. Under drought stress, maximum Gs was recorded in the relatively tolerant genotypes *Ca*74112 (60.25 ± 1.38 mmol m$^{-2}$s$^{-1}$), and *Ca*74110 (46.51 ± 0.89 mmol m$^{-2}$s$^{-1}$), and the minimum Gs in the sensitive genotypes *Ca*J-19 (36.84 ± 0.71 mmol m$^{-2}$s$^{-1}$) and *Ca*754 (30.28 ± 0.86 mmol m$^{-2}$s$^{-1}$) (Figure 7B). As a result of drought stress, the Gs decreased by 42.45%, 54.68%, 63.85%, and 70.24% in the genotype *Ca*74112, *Ca*74110, *Ca*J-19, and *Ca*754, respectively (Tables S12 and S18).

Under drought stress, at the end of the experiment, the higher transpiration rate was recorded in the relatively tolerant genotype *Ca*74112 (1.56 ± 0.07 mmol m$^{-2}$s$^{-1}$) and *Ca*74110 (1.11 ± 0.04 mmol m$^{-2}$s$^{-1}$), whereas the lower E value was displayed by the sensitive genotypes of *Ca*J-19 (0.54 ± 0.03 mmol m$^{-2}$s$^{-1}$) and *Ca*754 (0.42 ± 0.04 mmol m$^{-2}$s$^{-1}$) (Figure 7C). The minimum and maximum reduction of E as a result of drought stress were identified in the relatively tolerant genotypes of *Ca*74112 (63.6%) and *Ca*74110

(72.42%) and the sensitive genotypes of *Ca*J-19 (84.68%) and *Ca*754 (88.39%), respectively (Tables S13 and S18).

### 3.5.6. Photosynthetic Pigments

Plants grown under well-watered conditions had significantly higher pigment content than those in drought stress conditions. In drought stress, the result showed a significant decline of *Chl*-a (Figure 8A), *Chl*-b (Figure 8B), and total chlorophyll (Figure 8C) content in all tested genotypes whereas in well-watered plants, the amount of chlorophyll was relatively stable throughout the experiment. The highest and lowest *Chl*-a, *Chl*-b, and total chlorophyll content were detected in the relatively tolerant genotype *Ca*74112 (1.09, 0.41, and 1.5 mg g$^{-1}$fw, respectively) and *Ca*754 (0.63, 0.29, and 0.91 mg g$^{-1}$fw, respectively), respectively. The minimum and maximum reduction of *Chl*-a content as a result of drought stress were identified in the relatively tolerant genotypes *Ca*74112 (40.27%) and *Ca*74110 (47.28%) and the sensitive genotypes *Ca*J-19 (52.44%) and *Ca*754 (56.96%), respectively. The maximum reduction of *Chl*-b content was identified in the sensitive genotype of *Ca*754 (21.92%). Similarly, the minimum and maximum reduction of total chlorophyll content were identified in the relatively tolerant genotypes *Ca*74112 (35.57%) and *Ca*74110 (41.51%) and the sensitive genotypes *Ca*J-19 (45.49%) and *Ca*754 (49.94%), respectively.

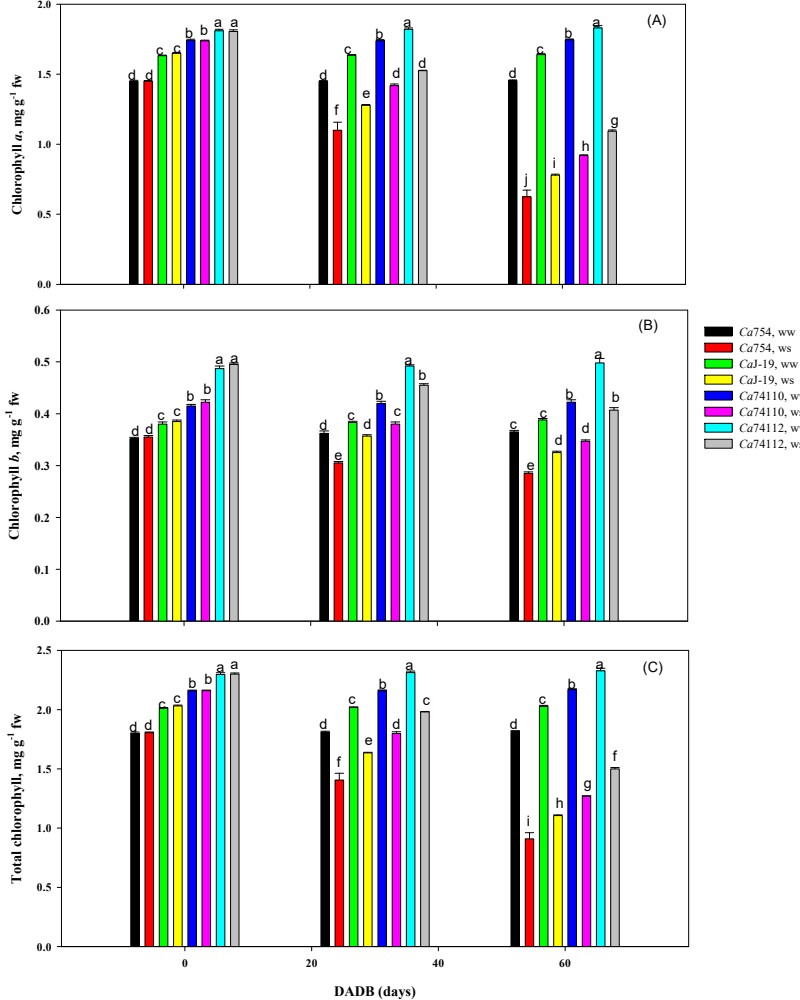

**Figure 8.** The effect of drought stress in (**A**) chlorophyll-a, (**B**) chlorophyll-b, and (**C**) total chlorophyll content of the four adult coffee genotypes, *Ca*754, *Ca*J-19, *Ca*74110, and *Ca*74112, under well-watered (*ww*) and drought stress conditions (*ws*). Bars indicate means ± SD (*n* = 15 replicates per genotype). Bars with the same letter indicate no significant difference at *p* < 0.05 between samples. DADB indicates the number of days after drought stress begins.

### 3.5.7. Cell Membrane Stability and Relative Cell Injury

Significant differences were observed for the mean cell membrane stability (CMS) and relative cell injury (RCI) under drought stress conditions among the genotypes, at the end of the experiment. The highest mean CMS, an indication of stress tolerance, was observed in the relatively tolerant genotypes of *Ca*74112 (82.5 ± 9.41%) and *Ca*74110 (73.31 ± 7.32%), and the lowest CMS in the sensitive genotypes *Ca*754 (49.94 ± 2.36%) and *Ca*J-19 (59.03 ± 2.81%). Inversely, the highest and lowest mean RCI, an indication of stress sensitivity, were observed in the sensitive genotypes *Ca*754 (50.06 ± 1.53%) and *Ca*J-19 (40.97 ± 1.47%) and the relatively tolerant genotypes *Ca*74112 (17.5 ± 4.48%) and *Ca*74110 (20.69 ± 3.41%), respectively (Figure 9).

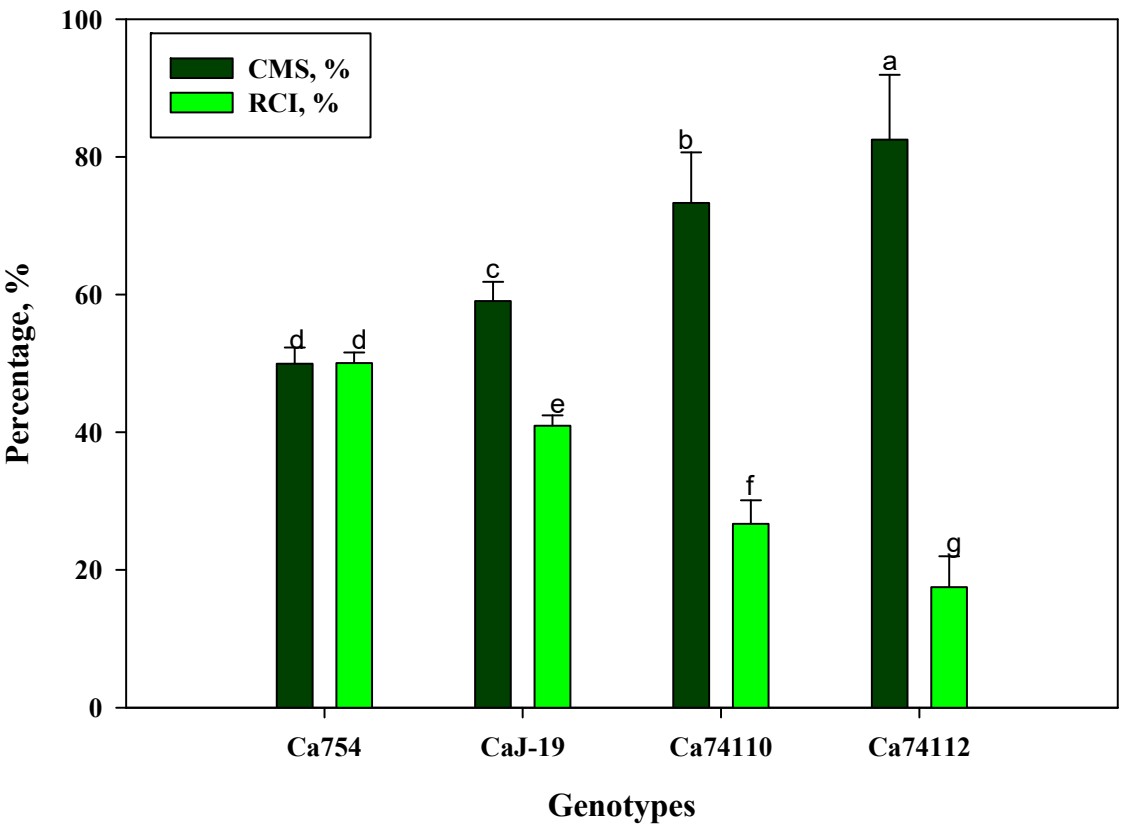

**Figure 9.** The CMS and RCI in the four adult coffee genotypes, *Ca*754, *Ca*J-19, *Ca*74110, and *Ca*74112, under well-watered (*ww)* and drought stress (*ws)* conditions, at the end of the 60 day experiment. Bars indicate means ± SD (*n* = 15 replicates per genotype). Bars with the same letter indicate no significant difference at *p* < 0.05 between samples.

### 3.6. Pearson Correlation Amongtested Parameters

Initial seed moisture content was positively and strongly correlated with seed length (*r* = 0.943), seed surface area (*r* = 0.94), and seed volume (*r* = 0.935), and negatively correlated with mean germination time (MGT) (*r* = −0.97), seed width (*r* = −0.969), and synchronization index (Z) (*r* = −0.978). Final germination percentage (GP) was positively and strongly correlated with pre-germination parameters, i.e., seed moisture content (*r* = 0.981), seed surface area (*r* = 0.987), seed length (*r* = 0.987), and seed volume (*r* = 0.985); with germination parameters, i.e., peak value (*r* = 0.995), Gv (*r* = 0.997), and MDG (*r* = 0.975); and post-germination parameters, i.e., seedling root length (*r* = 0.997), seedling shoot length (*r* = 0.98), seedlings root volume (*r* = 0.987), and R/S$^r$ (*r* = 0.951). However, GP was not significantly correlated with seed fresh weight (*r* = 0.168) and seed dry weight (*r* = −0.05), and negatively correlated with seed width (*r* = −0.99), mean germination time (*r* = −0.997), and Z (*r* = −0.997). The vigorous index of seedlings was also positively correlated with

initial seed moisture content ($r = 0.974$) and GP ($r = 0.99$), and negatively correlated with MGT ($r = -0.999$) and seed width ($r = -0.989$). In the adult coffee plants, under drought stress conditions, there was a strong positive correlation of stem $\Psi_w$ with relative water content ($r = 0.995$), stem length ($r = 0.949$), stem diameter ($r = 0.798$), leaf number ($r = 0.960$), leaf area ($r = 0.905$), root length ($r = 0.892$), root number ($r = 0.964$), root volume ($r = 0.827$), $CO_2$ assimilation rate ($r = 0.974$), stomatal conductance (Gs) ($r = 0.944$), transpiration rate (E) ($r = 0.973$), chlorophyll-a ($r = 0.950$), chlorophyll-b ($r = 0.903$), total chlorophyll ($r = 0.943$), and cell membrane stability ($r = 0.981$). Moreover, net assimilation rate (A) was highly correlated with relative water content ($r = 0.984$), leaf number ($r = 0.998$), stomatal conductance (Gs) ($r = 0.989$), and transpiration rate (E) ($r = 0.983$).

Furthermore, in order to understand the drought tolerance and sensitivity relationship between the seed quality traits and germination parameters, measurements of 90-day-old seedlings and adult coffee plants were established. It was shown that vital growth and physiological parameters of the adult coffee plants were positively correlated with key pre-, during-, and post-germination developmental events. The relative water content and stem water potential of the adult coffee plants were strongly correlated with pre-germination parameters such as seed moisture content ($r = 0.947$, $r = 0.966$, respectively), seed surface area ($r = 1$, $r = 0.994$, respectively), and seed volume ($r = 0.998$, $r = 0.994$, respectively); germination parameters such as germination percentage ($r = 0.989$, $r = 0.998$, respectively) and germination index ($r = 0.994$, $r = 0.999$, respectively); and post-germination parameters of 90-day-old seedlings such as stem length ($r = 0.974$, $r = 0.987$, respectively), root volume ($r = 0.976$, $r = 0.991$, respectively), and germinant vigorous index ($r = 0.991$, $r = 0.999$, respectively). Net assimilation rate was positively correlated with seed moisture content ($r = 0.958$), seed surface area ($r = 0.978$), germination percentage ($r = 0.975$), germination index ($r = 0.981$), and germinant vigorous index ($r = 0.972$). Chlorophyll-a and chlorophyll-b content was positively correlated with the vigorous index of 90-day-old seedlings ($r = 0.948$, $r = 0.898$, respectively). Cell membrane stability was positively correlated with seed moisture content ($r = 0.965$), seed surface area ($r = 0.983$), germination percentage ($r = 0.983$), germination index ($r = 0.988$), and vigorous index of 90-day-old seedlings ($r = 0.98$) (Table S14).

*3.7. PCA and Cluster Analysis*

The PCA analysis was performed on a dataset containing measurements of seed quality traits, germination parameters, growth and development parameters of 90-day-old seedlings, and adult coffee plant measurements. The eigenvalues and loading contribution rates of principal components were the basis for selecting principal components. Two principal components were obtained, and their contribution rates were PC1 (90.61%) and PC2 (7.23%), respectively, with a cumulative contribution rate of 97.84% (Figure 10, Table S15). Therefore, the first two principal components were selected as the important principal components of the drought tolerance and sensitivity responses of the coffee genotypes. Apart from seed width, mean germination time, and synchronization index (Z), all other parameters positively and strongly contributed to PC1, while seed fresh and dry weight highly and positively contributed to PC2 (Figure S9, Table S16). To assess the drought tolerance and sensitivity responses of the genotypes based on all the parameters mentioned above, the score value of PC1 and PC2 to the sum of the total PC values of the extracted principal components was taken as the weight. The higher PC score values were positive and correlated with the drought-tolerant genotypes of *Ca*74112 (8.00 PCA score value) and *Ca*74110 (1.70 PCA score value), whereas the lower PC score values were identified in the sensitive genotypes of *Ca*J-19 ($-1.40$ PCA score value) and *Ca*754 ($-8.30$ PCA score value) (Figure S10, Table S17).

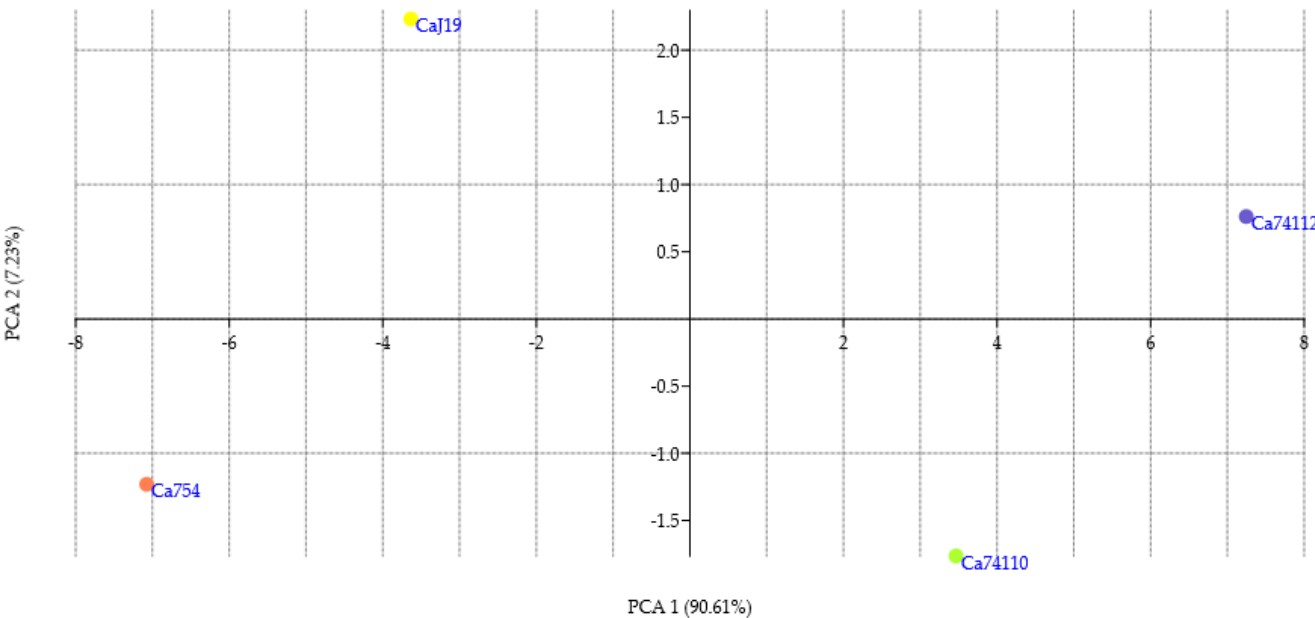

**Figure 10.** Principal component analysis based on variables presented in this study (full list of parameters presented in Table S16) of traits of seed, germination events, 90-day-old seedlings, and adult coffee plants.

Based on the Euclidean and Manhattan similarity index analyses, using seed quality traits, germination parameters, growth and development parameters of 90-day-old seedlings, and measurements of adult coffee plants, both the tolerant genotypes (*Ca*74112 and *Ca*74110) were highly related and form one category, and the sensitive genotypes (*Ca*J-19 and *Ca*754) were also highly related and form the other category (Figure S11).

## 4. Discussion

### 4.1. Seed Trait Variation Associated with Germination Potential

The size and weight of the coffee seeds are usually influenced by both internal (i.e., genetic makeup, hormones, water content) and external factors (i.e., available water, storage techniques, etc.). Both seed size and weight are correlated with the initial seed moisture content, surface area, volume, and germination percentage [13,21]. One of the key seed traits associated with germination is the moisture content, which is vital for determining the germination potential of coffee seeds [18,59]. Seeds with higher moisture content provide efficient germination capacity and improve the seed's potential to tolerate drought conditions [17]. In the current study, seeds of relatively tolerant genotypes retain significantly higher moisture content (i.e., *Ca*74140 14.89%, *Ca*74112 14.71%, and *Ca*74110 14.58%) than the other genotypes. Sensitive genotypes (*Ca*754 8.05%, *Ca*J-19 10.87%, and *Ca*Geisha 11.38%), on the other hand, retained the lowest seed moisture content which may be associated with their sensitivity to drought. A coffee seed with moisture content lower than 9% is characterized by having poor germination performance and high sensitivity to drought. Seeds with moisture content between 9 and 13% exhibit an improved germination potential but are still moderately sensitive to drought. When the moisture content is above 13%, the seeds have much more reliable germination potential and are known to be drought-tolerant [17,29].

Relatively drought-tolerant genotypes (*Ca*74140, *Ca*74112, and *Ca*74110) tend to have more elongated seeds than the sensitive genotypes in our study. Elongated seeds usually have a higher surface area-to-volume ratio. Previous studies found a positive relationship between seed length and germination parameters [60]. Seeds with a higher surface area-to-volume ratio (i.e., relatively tolerant genotypes in this study) could absorb more water from the soil and germinate more quickly than those with a lower surface area-to-volume

ratio [61]. Elongated seeds with a high surface area-to-volume ratio will have more contact with the soil moisture and air, which increases the potential of seeds to imbibe more water molecules so that they will have high germination potential. Meanwhile, genotypes with relatively heavier seed weight (e.g., *Ca*74140 and *Ca*74112) supposedly have more reserve foods (cellulose, hemicellulose, and insoluble mannan) inside the endosperm and potentially are more tolerant to environmental stress conditions [21,62,63]. Similar to the previous findings [43,64], our study affirms that genotypes having elongated seeds with higher surface area to volume ratio and higher mass such as the relatively tolerant genotypes *Ca*74140, *Ca*74112, and *Ca*74110 are characterized by efficient germination even under drought stress conditions.

### 4.2. Speed of Morphological Changes during Germination Is Highly Associated with Genotypes

Slow and asynchronous germination of coffee seeds is in part due to the differences in the imbibition of seeds during the germination process [19,21]. Hydrophilic molecules found in the outer and harder part of the coffee endosperm seed coat facilitate water absorption and cause the seed to become turgid and rounded [10,21]. Furthermore, the endosperm wall stretches, resulting in structural changes in seeds in all dimensions. Some research also noted that coffee seeds with higher content of hydrophilic molecules in the endosperm get hydrated and germinate faster even under limited water availability or in drought stress conditions [13]. In the present study, the relatively tolerant genotypes (i.e., *Ca*74112, *Ca*74110, and *Ca*74140) had the shortest hydration time, and are likely to contain more hydrophilic compounds, enabling them to germinate quickly and withstand water shortages. Studies have suggested that faster hydration of coffee seeds allows more oxygen to enter the embryo and activates aerobic respiration [15,17,20]. This process triggers the activation of hydrolyzing enzymes that catalyze the hydrolysis of food reserves in the endosperm [16,17,65], resulting in the transformation of the quiescent embryo to a metabolically active one [15,62,66,67]. Other experiments also suggested that the difference in the rate of imbibition and hydration has a direct effect on the germination speed and the growth rate of coffee seedlings [14,68].

The initial germination events such as water imbibition, $O_2$ entry, sub-cellular structural changes, molecular synthesis, and cellular respiration lead to cell division [17,21], and radicle development that breaks through the outer layer of the endosperm [22,64]. The fastest germination observed in the relatively tolerant genotypes (*Ca*74112, *Ca*74110, and *Ca*74140) could be attributed to a shorter period of initial germination events. The higher value of mean germination time is correlated with rapid seed germination even under drought stress conditions and with high tolerance capacity during the early germination period [13,27,66]. Consequently, relatively tolerant genotypes completed the subsequent post-germination events (*sd*-1 to *sd*-5) earlier than the other genotypes. The early completion of germination events in these genotypes may allow the seedling to grow rapidly and survive in a resource-limited environment [10,22].

### 4.3. Germination Performance Variability Is Highly Related to Genotype

Germination percentage is a widely used parameter to predict the potential for germination and seedling establishment of a given lot of seeds [67,69,70]. The higher the germination percentage value, the greater the germination of a seed population [62,66,71]. Seeds with large surface area and volume are characterized by having an improved cellular division, elongation, differentiation, and growth that lead the seeds to attain high germination percentage [13,21,66]. In the current study, the germination percentage was higher in relatively tolerant genotypes and strongly correlated with the seed's surface area ($r = 0.960$), volume ($r = 0.954$), and seed length ($r = 0.918$). For example, genotype *Ca*74140, with the highest moisture content, surface area, and volume exhibited the highest germination percentage ($90 \pm 1.44\%$). Such genotypes are characterized by the potential to produce seedlings that can tolerate stress conditions [17,72].

The mean germination time indicates the average duration of time required for the utmost seed germination performance, i.e., lower values of mean germination time indicate faster germination [11]. This is directly related to the seed volume, i.e., the higher the volume of seeds, the lower the mean germination time [72]. For example, in this study, relatively drought-tolerant genotypes (*Ca*74140, *Ca*74110, and *Ca*74112) possess both higher moisture content and significantly lower mean germination time compared with the sensitive genotypes. The high volume of endosperm promotes faster germination, i.e., lower mean germination time [12]. This is because the presence of a large food reserve allows for efficient metabolic activities, thus shortening the germination period and granting the capacity to grow even under drought stress conditions [15,17]. In this study, the coefficient of the velocity of germination and coefficient of variation of germination time were markedly higher in the relative drought-tolerant genotypes (*Ca*74112, *Ca*74140, and *Ca*74110) compared to the sensitive (*Ca*754, *Ca*J-19, and *Ca*Geisha) genotypes, indicating that germination was rapid but spread out over time in the tolerant genotypes. Additionally, genotypes *Ca*74112, *Ca*74140, and *Ca*74110 exhibited a higher germination rate index and germination index values, which are characteristics of drought-tolerant behavior [72].

The mean number of seeds germinated per day (MDG) is directly associated and strongly correlated ($r = 0.96$) with the final germination percentage. The peak value (Pv) of germination (the maximum cumulative germination percentage per the number of total days) and mean daily germination can be counter balanced, resulting in equal values for germination value (Gv) for samples or treatments with different behavior concerning the germination process [73]. Genotypes with higher Gv, such as *Ca*74112, *Ca*74110, and *Ca*74140, have a high mean germination time and rapid vegetative growth, which can increase their capacity to withstand drought stress conditions [42]. The intrinsic traits of seed size and weight had a strong effect on mean germination time and germination percentage, which corroborates the relatively tolerant genotypes (*Ca*74140, *Ca*7412, and *Ca*74110). Moreover, the relatively tolerant genotypes display more vigorous growth and development of roots (high root branching root length, root number, accompanied by dense root hairs, etc.) and shoots. In addition, a root-to-shoot ratio greater than 1 in relatively tolerant coffee genotypes (i.e., longer root than hypocotyls) is an indication of stable root to shoot balance. Other research found that, an early coffee seedling with such a shoot and root structure results in the plant developing higher shoot and root surface area and biomass. Well-developed roots provide efficient mechanical anchorage in the soil, improve the rate of nutrient and water uptake, boost photosynthetic activity, and promote seedlings' plasticity to adapt to various environmental conditions, including drought stress [4,74]. In addition, some authors have concluded that the development of such quality of root and shoot at early stages of development, as in the genotypes *Ca*74140, *Ca*74112, and *Ca*74110, has a far-reaching influence on withstanding drought stress and thus improving the growth and yield of adult coffee plants [11,13,17,21].

*4.4. The Extent of Drought Stress Impact on Growth Variesamong Coffee Genotypes*

Drought stress is well known to reduce the growth and physiological processes of *C. arabica* [59,75]. Under drought, there is a decline in turgor pressure that leads to a reduction in cell division, elongation, and expansion, which then decreases growth and development, gas exchange, and morphological, molecular, and other biochemical activities [76]. In this study, the result showed that the impact of drought stress varies among the genotypes, and yet it reduced the growth of shoot, root, biomass, water relations, gas exchange, chlorophyll pigments, and cell membrane stability, and increased stomatal densities and relative cell injury.

Our results revealed that the relatively tolerant genotypes of *Ca*74110 and *Ca*74112 showed much better efficiency in terms of key shoot growth and developmental indicators such as height, collar diameter, leaf area, and leaf number than the sensitive genotypes of *Ca*754 and *Ca*J-19. Shao et al. [77] stated that the impact of drought stress usually depends on the intensity, severity, duration, genotype, and growth stage of the plant. According

to Oguz et al. [78] and Cai et al. [79], a shortage of water content or turgor pressure disrupts the cellular mitosis process that greatly restricts and reduces cell division, cell elongation, and differentiation, and consequently limits the growth and development of shoots and roots. Similar to this study, Tounekti et al. [80] reported that the minimum and maximum growth reduction impact of drought stresses in stem height and diameter and leaf number and area are caused by suppressing cell division and elongation. Tavares-Junior et al. [53] and Gheidary et al. [81] reported the reduction of shoot height, collar diameter, leaf area, and disruption and abortion of buds and flowers under drought stress conditions. Findings from Razmjoo et al. [82] stated that drought stress reduced the entry of macro- and micronutrients into the plant, leading to a reduction in shoot length. The development of sufficient leaf number and leaf area is vital for the coffee plants for effective photosynthesis, which has a great impact on the growth and development [83]. The lower the leaf area and number, the lower the photosynthetic rate, and the fewer the catabolic and anabolic biochemical reactions to supply the molecules needed for the growth and development of a plant [29]. Similar to this study, Bhargavi et al. [84] reported a reduction in the number of leaves when *Andrographis paniculate* was subjected to drought stress. Srivastava and Srivastava [85] also reported the declining leaf area in *Petroselinum crispum* L. and *Stevia rabaudiana* when grown under drought stress conditions. Furthermore, Shao et al. [77] also stated that, apart from the reduction in the number of leaves and leaf sizes, an increase in leaf senescence is a consequence of drought stress.

Roots play a key role in the acquisition of water and nutrients, provide structural support, ensure tolerance against abiotic stresses, and regulate rhizosphere zone and absorption by symbiotic associations with other microorganisms [26,86]. In the present study, drought stress influenced the growth and development of the root system in plants belonging to the four coffee genotypes, but the impact varies among the genotypes. The reduction in root growth is attributed to a decrease in root number and length, which ultimately decreased the volume of the root system. Under drought stress, the relatively tolerant genotypes *Ca*74112 and *Ca*74110 displayed greater root length, number, and volume, compared with the sensitive genotypes of *Ca*754 and *Ca*J-19.According to Wright et al. [87], roots are the first signaling parts of the plant to recognize the availability of soil water content, which then promotes the adjustment of the root's growth and development in terms of root type, length, number of lateral roots, volume growth, and organization characteristics. Tagliavini et al. [88] and Hussain et al. [89] reported that drought-tolerant genotypes develop deeper and more developed root systems to support the plants in acclimatizing to a wide range of drought stress conditions. When drought-tolerant plants are grown under water deficit conditions, developmental changes and maximization of root number, length, density, volume, size, and diameter are more evident than for drought-sensitive plants [27,75]. Osakabe et al. [90] and Smith and De Smet [91] stated that the elongation of the root is a vital strategy to maximize the retention of soil water content, and nutrient absorption to improve the plant root-to-shoot proportion, and subsequently reducing the plant biomass. Under drought stress, those genotypes with long and advanced root systems can easily uptake water and nutrients and have successful plant growth and development by avoiding drought stress [92–94].

As a result of the reduction of shoot and root growth and development under drought stress conditions, the reduction of fresh biomass and dry biomass was observed but in the relatively tolerant genotypes (*Ca*74112 and *Ca*74110), mass decrease was significantly lower than the sensitive genotypes (*Ca*754 and *Ca*J-19). The reduction of biomass under drought stress conditions in four of the genotypes was in line with the results obtained by Dias et al. [75] and Poorter et al. [95].

DaMatta and Ramalho [35] and DaMatta [96] reported the reduction of fresh and consequently dry weight in coffee plants when they were subjected to drought stress. Dias et al. [75] reported a decrease in dry mass in drought-tolerant (Siriema) and drought-sensitive (Catucai) *C. arabica* genotypes under drought. Similar to our study, the level of mass reduction was smaller for drought-tolerant plants than for drought-sensitive ones.

### 4.5. Relative Water Contentand Water Potential Affected Differently among Genotypes under Drought Stress

The relative water content is a good reference for the water conditions of the plant as it represents the balance between water supply and transpiration [26,97,98]. Drought stress leads to a decrease in plants' relative water content and water potential [27,99]. The result of the current study revealed that under control conditions all plants regardless of their genotype possessed significantly similar relative water content throughout the experiment ranging between 80 and 84% and decreasing to 30–49% under drought stress treatment. Moreover, under drought stress conditions, the highest relative water content was recorded by the relatively tolerant genotype *Ca*74112 (41.89% reduction), while the lowest was recorded in the sensitive genotype *Ca*J-19 (60.32% reduction). In this study, under drought stresses, the decrease in water potential depends on the genotype, i.e., a more significant decline was observed for drought-sensitive genotypes.

Based on Barrs and Weatherley [55], leaves with relative water content between 30 and 40% reveal plants under water scarcity. According to Silva et al. [100], when relative water content values are around 98%, it refers to leaves' turgidity. It is known that smaller reductions in water potential correlate with higher water retention capacity and adaptation to drought stress [98]. The difference in the decline of water status among the four genotypes is related to their unique stomatal control and cuticular transpiration rate [35]. The results presented in this study indicate that the relatively tolerant genotypes of*Ca*74112 and *Ca*74110 can maintain a higher water balance than the sensitive genotype*Ca*J-19 and *Ca*754. The leaves of *Ca*74112 and *Ca*74110 were waxier than the leaves from the other genotypes, which would be beneficial for reducing water transpiration during drought stress. Similar to this study, Soltys-Kalina et al. [101] reported the reduction of the relative water content of potatoes under drought stress conditions, but the report also identified that the tolerant genotypes are characterized by having higher relative water content than the sensitive genotypes. Genotypic variation of water potential may be attributed to differences in the ability to absorb more water from the soil and the ability to reduce water loss through stomata [102]. It may also be due to differences in the ability of genotypes to maintain tissue turgor and hence physiological activities [100,103]. The results herein were similar to those from studies performed in *C. arabica* by Dias et al. [75] and Tounekti et al. [80], where sensitive genotypes such as *Ca*J-19 and *Ca*754 with lower relative water content and water potential also had low photosynthetic potential, which confirmed the lower efficiency of $CO_2$ assimilation, which in turn resulted in lower growth performance under drought stress due to water limitation. The genotype having higher root length and volume, such as the relatively tolerant genotypes *Ca*74112 and *Ca*74110, under drought stress, should have high relative water content and water potential [104,105].

### 4.6. Drought Stress Affected the Gas Exchange Differently in Coffee Genotypes

Gas exchange parameters such as net photosynthesis rate, transpiration rate, and stomatal conductance are key indicators of water shortage in plants and are useful to evaluate the tolerance responses of genotypes [96,106,107]. With the increasing intensity of drought stress, plants are subjected to the reduction of photosynthesis assimilation rate, stomatal conductance, and transpiration rate by influencing the indices of gaseous exchange parameters [27,29,108,109].

As observed in this study, there was a strong relationship between the water potential reduction in the genotypes, and the responses of net photosynthesis rate, transpiration rate, and stomatal conductance. Those parameters were stable in control conditions and decreased as drought stress was prolonged regardless of the genotype. As the drought stress intensified, the $CO_2$ assimilation rate in the sensitive genotypes (*Ca*754 and *Ca*J-19) decreased more than in the relatively tolerant genotype (*Ca*74112 and *Ca*74110).

The lower water availability in plants usually leads to the reduction of water in the root–stem–leaf continuum [26,110–112]. Water deficit leads to the reduction of photolysis reaction in the photosystem that minimizes the formation of free hydrogen ions and

electrons during electron transport chain systems, that in turn inhibits adenosine triphosphate (ATP) and nicotinamide adenine dinucleotide phosphate (NADPH) production, which are utilized in the dark reaction of photosynthesis [113,114].In addition, drought stress-induced abscisic acid (ABA) promotes stomatal closure to conserve the remaining internal water from loss and consequently lowering the internal $CO_2$ concentrations in the mesophyll and decreasing $CO_2$ fixation by inhibiting the synthesis of ribulose bisphosphate (RUBP) [27,115,116].

In this study, stomatal conductance was also strongly affected by the changes in the leaf water potential. As the drought stress intensified, the decrease in stomatal conductance with decreasing leaf water potential was more substantial in *Ca*754 and *Ca*J-19 than in *Ca*74112 and *Ca*74110, indicating that the sensitivity of stomata to low leaf water potential conditions was higher. Similarly, the result of this study also identified the reduction of transpiration rate under drought stress conditions. The relatively tolerant genotypes showed a reduction of E by 63.6–72.42%, whereas higher reductions were recorded in the sensitive genotypes, i.e., between 84.68% and 88.39%. Similar to our findings, Dias et al. [75], DaMatta et al. [117], and Silva et al. [100] reported that relatively tolerant coffee genotypes have the potential to display much-improved photosynthesis assimilation rate, stomatal conductance, and transpiration rate, unlike the sensitive genotypes, even under drought stress conditions.

### 4.7. Impact of Drought Stress on Photosynthetic Pigments Concentration

In the present study, at the beginning of the stress experiment, Chl-a, Chl-b, and total chlorophyll contents showed no significant difference among the genotypes, but as the drought stress intensified, it was found to cause pronounced reductions and was strongly correlated with the reduction of relative water content and water potential. The relatively tolerant genotypes of *Ca*74112 and *Ca*74110 showed less reduction in Chl-a, Chl-b, and total chlorophyll contents than the sensitive genotypes of *Ca*J-19 and *Ca*754. These reductions might be attributed to reduced water supply and a decrease in leaf water content, which restricts the movement of nutrients responsible for the synthesis of pigments, which ultimately declined the synthesis of photosynthetic pigments [110,118]. Drought stress also influences the structure and functions of photosynthetic pigments by damaging the complex protein structures of the thylakoid membranes and decreasing the activities of RUBISCO enzymes [26,29]. The reduction is associated with the damage that occurred in the pigments of the light-harvesting complex proteins, which directly impacts the photon absorption and electron transport chain [100,111,119]. Jaleel et al. [120] also reported that intensified drought stress causes the degradation of photosynthetic pigments, damage to the membrane system, and the reduction of synthetase activity. Kirnak et al. [121] associated the reduction of chlorophyll concentrations with increased electrolyte leakage due to softening and breakage of the cell wall. Studies reported by Manivannan et al. [122], Nikolaeva et al. [113], and Mafakheri et al. [123], indicated that with increased drought stress, leaf chlorophyll content showed a rapid decline in many crops, such as wheat, chickpea, and sunflower.

### 4.8. The Magnitude of Cell Membrane Stability and Relative Cell Injury Differ among Genotypes under Drought Stress

Since the amount of electrolyte leakage is a function of membrane permeability, the degree of cell damage resulting from water deficit can be assessed by measurement of electrolytic conductance [124]. Moreover, cell membrane stability (CMS) and relative cell injury (RCI) in response to drought stress were confirmed to be an important measure of tolerance and sensitivity to be used to screen for genetic variation in drought tolerance among species and genotypes [27,125]. The result of this study revealed that drought stress had a significant impact on CMS in the four genotypes under investigation. The relatively tolerant genotype *Ca*74112 showed a strong CMS of 82.5%, which was followed by *Ca*74110 with CMS of 73.31%, indicating a high level of drought stress tolerance response. Signif-

icantly lower values were displayed by the sensitive genotypes, i.e., *Ca*J19—59.03% and *Ca*754—49.94%, indicating relative sensitivity to drought stress. In addition, the relationship between CMS and RCI is inversely proportional and strongly negatively correlated.

It was shown that drought-tolerant varieties have higher membrane stability under drought stress than drought-susceptible varieties [126,127]. In response to drought stress, the level of membrane stabilizers, i.e., heat-shock proteins, and saturated lipids, increased in drought stress-tolerant genotypes to maintain the membrane permeability and integrity [128,129]. Prasch and Sonnewald [130] and Zhou et al. [131] also reported that, as a result of membrane leakage, photosynthetic products such as sugars explicitly accumulated during drought stress conditions.

*4.9. Multivariate Analysis of Parameters Analyzed in this Study*

The result showed that, under drought stress conditions, the main differences among the genotypes were recorded in the case of adult coffee vegetative growth, gas exchange, water parameters, stomatal densities, and cell membrane stabilities. The changes in the growth and physiological parameters of the coffee genotypes varied under drought stress, and the responses were different in different periods, which indicated that genotypes have different ways to adapt to drought stress. The PCA results showed that, apart from seed width, mean germination time, and synchronization index (Z), all other parameters positively and strongly contributed to PC1. During continuous drought stress, plants experience stress, adaptation, injury, and repair. The comprehensive adjustment of different response mechanisms in different stress stages constitutes the overall drought resistance of plants [80]. According to de Oliveira et al. [132], plants that have the same drought tolerance belong to the same PCA group category and their stress response is similar. Moreover, their clustering based on the Euclidean and Manhattan similarity indexes also rest in the same group. In this study, the response to drought of the relatively tolerant genotypes (*Ca*74112 and *Ca*74110) and sensitive genotypes (*Ca*J-19 and *Ca*754) was similar, which was confirmed by PCA and cluster analysis.

**5. Conclusions**

Drought stresses had detrimental effects on germination, seedling growth, and the morphological and physiological performance of adult plants, as evident by the changes observed for all the studied traits. The present study highlighted significant differences in germination duration and post-germination growth stages among coffee plants of different genotypes. The relatively tolerant genotypes exhibited faster completion of each germination and seedling development stage, resulting in stronger seedlings compared to moderately sensitive and sensitive groups. The seed's inherent trait played a crucial role in germination and was reflected in the post-germination development, suggesting that seed trait analysis could enhance coffee seed germination and crop yield. Notably, the relatively tolerant genotypes (*Ca*74112 and *Ca*74110) demonstrated superior growth performance against drought stress compared to other genotypes.

Based on these findings, further research is recommended to investigate seed priming and microbial inoculations as potential methods to boost the slow germination process in *C. arabica*. Additionally, in order to delve further into the mechanisms of drought stress tolerance in arabica coffee genotypes, it is imperative to explore the role of membrane stabilizing factors, metabolic profiling, proteins (aquaporins) related to water transport, and gene expression studies.

**Supplementary Materials:** The following supporting information can be downloaded at: https://www.mdpi.com/article/10.3390/agriculture13091754/s1, Figure S1. Seeds of the nine *C. arabica* genotypes used in this study; Figure S2. Microphotographs of: (A) basal surface of coffee seed endosperm, without exocarp, mesocarp, and endocarp, (B) cross-section of a *C. arabica* seed showing the folding of the endosperm and embryo localization, and (C) profile section of coffee berry and bean anatomy, including the pericarp, mesocarp, endocarp, spermoderm, and endoderm. Observations for A and B were conducted under a Leica MZ8 microscope with a resolution power of 100 dpi.; Figure S3. The shade

and greenhouse for the germination of *C. arabica* seeds: (A) washed and autoclave-sterilized sand arranged for sowing the coffee seeds in a plastic tray with the hole at the base, (B) poly-propagator that provides efficient microclimatic conditions for the germination of the coffee seeds, and (C) upward growth of germinant, during the study period; Figure S4. Representative example of *C. arabica*s and germination process of the nine genotype seeds: (A) early stage (maximum $26.0 \pm 2.31$ days), (B) matchstick stage (max. $32.0 \pm 2.09$ days), (C) butterfly stage (max. $46.0 \pm 2.23$ days), and (D) transplanting stage (max. $53.2 \pm 3.86$ days); Figure S5. Transplanting the *C. arabica* genotypes (A) from the sand media, (B) pulling up the genotypes without root damage, (C) the initial seedling, (D) initial seedling after transplanting, (E) seedlings at the age of 6 leaf pairs (6monthsold), (F) at the start of the experiment when genotypes developed 7–8 leaf pairs (8–9 months old); Figure S6. Comparing the shoot growth differences of *C. arabica* genotypes growing under well-watered (ww) and drought stress (ws) conditions (after 60 days of drought treatment); Figure S7. Comparing the root growth differences of *C.arabica* genotypes growing under well-watered (ww) and drought stress (ws) conditions (after 60 days of drought treatment); Figure S8. Comparing the biomass of *C. arabica* genotypes (A) growing under well-watered conditions and (B) growing under drought stress conditions (after 60 days after drought stress treatment); Figure S9. PC1 and PC2 loading and correlation plot with the various stress indicator parameters of the four coffee genotypes, *Ca*754, *Ca*J-19, *Ca*74110, and *Ca*74112, under ws conditions; Figure S10. PCA score value and the cumulative contribution rate of PC1 and PC2 of variables tested in this study; Figure S11. (A) Hierarchical clustering using Euclidean similarity index, and (B) neighbor joining clustering using Manhattan similarity indexes of selected traits of seed, germination events, 90-day-old seedlings, and adult coffee plants. Table S1. Parameters tested in this study; Table S2. Mean values and SD of pre-germination parameters; Table S3. Mean values and SD for germination parameters; Table S4. Mean values and SD of stem height (cm); Table S5. Mean values and SD of stem diameter (mm); Table S6. Mean values and SD of leaf number; Table S7. Mean values and SD of leaf area ($cm^2$); Table S8. Mean values and SD of root length (cm), root number, and root volume ($cm^3$); Table S9. Mean values and SD of relative water content (%); Table S10. Mean values and SD of stem water potential ($\Psi$w, $-$Mpa); Table S11. Mean values and SD of net assimilation rate (A, $\mu mol\ m^{-2}s^{-1}$); Table S12. Mean values and SD of stomatal conductance (Gs, $mmol\ m^{-2}s^{-1}$); Table S13. Mean values and SD of transpiration rate (E, $mmol\ m^{-2}s^{-1}$); Table S14. Pearson correlation analysis and heat-map of seeds, germination events, germinant, and adult coffee genotypes of the four coffee genotypes, *Ca*754, *Ca*J-19, *Ca*74110, and *Ca*74112, under drought stress conditions; Table S15. PCA eigenvalues; Table S16. PCA Loading contribution; Table S17. PCA Score value for each coffee genotype; Table S18. Two-way ANOVA.

**Author Contributions:** Conceptualization, A.D., B.W. and H.C.; methodology, A.D., T.S. and H.C.; investigation, H.C. and Y.B.; writing—original draft, H.C.; writing—review and editing of the manuscript, H.C., A.D., G.B.D. and A.M.-A.; visualization, A.D., H.C. and A.M.-A.; funding acquisition, A.D., B.W. and G.B.D.; resources, A.D., B.W. and H.C.; supervision, A.D., T.S., B.W., G.B.D. and A.M.-A. All authors have read and agreed to the published version of the manuscript.

**Funding:** We thank Addis Ababa University for providing the research fund (LT/PY-038/2019) and the Department of Plant Biology and Biodiversity Management for facilitating the financial support.

**Institutional Review Board Statement:** Not applicable.

**Data Availability Statement:** Relevant data applicable to this research are within the paper and are also available on request from the corresponding author.

**Acknowledgments:** We are grateful to the Jimma Agricultural Research Center for providing the seeds of coffee (*C. arabica*) genotypes.

**Conflicts of Interest:** The authors declare no conflict of interest.

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
