# Peer review of "Unraveling Drought Tolerance and Sensitivity in Coffee Genotypes: Insights from Seed Traits, Germination, and Growth-Physiological Responses"

_agriculture, doi:10.3390/agriculture13091754_

Round 1
Reviewer 1 Report
Though the MS is in good form, there is still room for further improvement.
Without line numbering, it is always hard to write line-by-line comments.
At the start of the abstract, please add 1-2 lines justifying the need for current study.
In the introduction, authors must use some of the most recent studies, such as doi: 10.1002/tpg2.20279; 10.1002/ggn2.202100017.
In the results, please do not simply write what was observed in the figures. Authors should also tell us what the data reflect.
Add more information about the traits associated with drought tolerance and sensitivity.
Minor editing of English language required
Reviewer 2 Report
This work study the correlation of seed traits with drought tolerance of different coffee genotypes, as a way to find a connection that could assist selection of genotypes with higher drought tolerance and yield potential. Experiments were performed with seeds from 9 coffee genotypes, and very little information is given about conditions of seed production or how seeds were handled before evaluation. I think this is an important problem, because it is well known that seed quality and performance depend not only on their genotype, but importantly on the environmental conditions before and after the harvest. Because the clear differences observed in seed germination among seedlots (Figure 4a), resulted evident and predictable that vigor and performance of seed and seedlings would be different among the genotypes, which could be due to genotype effect or simply to differences in initial seed quality (a seed with low germination is a seed with low vigor, which mean low germination rates, low seedling growth rates, low uniformity, low resistant to environmental stresses, etc). I think that a requirement for a study like this would be to star with seeds of similar physiological quality (at least a minimal standard germination, as required for seed commercialization). Additionally, more than a seed lot of each genotype should be used, so you can assume that observed differences are due to genotype.
Other problems of the study are:
Very low number of evaluated seeds (3 reps of 20 seeds, while in commercial seed evaluations 4 reps of 100 seeds re required).
Seed water content (SWC) depends of seed genotype, particular seed characteristics of each seed lot (seed size, chemical seed content, …) and environment where seeds are stored (specially relative humidity). If authors want to prove that SWC of seeds from different genotypes is different, they should ensure to keep the seeds under a common and controlled environment (RH) for some time before SWC evaluation. In addition, different seedlots of each genotype should be evaluated.
It is not clear the number of replications used in the growth and physiological experiments. It says that “The experimental design was a completely randomized block design, forming a 4 * 2 factorial (four genotypes and two water applications), with a replication of 15 genotypes, comprising a total of 120 coffee plants”, so 15 replications, one plant per replication, is assumed. However, there are measures that are destructive with plants stems or leaves (stem water potential, leaf relative water content) and are performed each 10 days, during 60 days…so it is not clear how many plants were used for each measurement.
Reviewer 3 Report
Congratulations to the authors for presenting a comprehensive and clear manuscript on germination in coffee seedlings. The importance of genotype is an interesting feature highlighted in this study, so the role of breeding to develop higher yielding and stress tolerant coffee genotypes could be emphasised more in the introduction and discussion. Please find my major and minor points discussed below.
Major points:
The manuscript is quite long so the authors could try to move less important figures and text to the supplementary in order to aid the reader in following the narrative of the manuscript.
The statistics are well described, however I wondered about the normality of the data and the underlying distributions. For example, Pearson correlations should only be performed on normally distributed data, whereas other correlation methods are available for non-normally distributed data (e.g. Kendall correlation). Could the authors clarify whether their data were normally distributed in the methods section? If the data is not normally distributed, then alternative statistical methods need to be employed other than Pearson's correlation.
Another aspect of the statistical analysis that would be useful to include is the results of the ANOVAs as an ANOVA table. There is a great amount of detail when it comes to comparing individual cultivars (through Tukey multiple comparisons), however the original ANOVA analysis would help the reader to see the significance of the main experimental variables (genotype and water treatment). This would also help to assess whether there is an interaction between genotype and water treatment (i.e. whether different genotypes respond differently to water stress). Inclusion of this in the paper would be very useful.
Minor points:
Table 1: " 5-photosynthetic cotyledons" - this should be 5 photosynthetic leaves not cotyledons (coffee is a dicot - it only has 2 cotyledons).
Figure 9: I don't think its appropriate to have a line connecting these points as these data are different cultivars and not connected in any meaningful way.
Very clear and easy to read. Some minor spell-checking required (e.g. "dermintal" in the Conclusion should be "detrimental").
Round 2
Reviewer 1 Report
The revised version can be accepted
Minor editing of English language required
Author Response
Dear Reviewer,
Please see the attachment.
Regards,
